# ZPressor: Bottleneck-Aware Compression for Scalable Feed-Forward 3DGS

**Weijie Wang**[1]    **Donny Y. Chen**[2]    **Zeyu Zhang**[2]
**Duochao Shi**[1]    **Akide Liu**[2]    **Bohan Zhuang**[1]

[1]ZIP Lab, Zhejiang University    [2]Monash University

## Abstract

Feed-forward 3D Gaussian Splatting (3DGS) models have recently emerged as a promising solution for novel view synthesis, enabling one-pass inference without the need for per-scene 3DGS optimization. However, their scalability is fundamentally constrained by the limited capacity of their models, leading to degraded performance or excessive memory consumption as the number of input views increases. In this work, we analyze feed-forward 3DGS frameworks through the lens of the Information Bottleneck principle and introduce ZPressor, a lightweight *architecture-agnostic module* that enables efficient compression of multi-view inputs into a compact latent state $Z$ that retains essential scene information while discarding redundancy. Concretely, ZPressor enables existing feed-forward 3DGS models to scale to over 100 input views at 480P resolution on an 80GB GPU, by partitioning the views into anchor and support sets and using cross attention to compress the information from the support views into anchor views, forming the compressed latent state $Z$. We show that integrating ZPressor into several state-of-the-art feed-forward 3DGS models consistently improves performance under moderate input views and enhances robustness under dense view settings on two large-scale benchmarks DL3DV-10K and RealEstate10K. The video results, code and trained models are available on our project page: https://lhmd.top/zpressor.

## 1 Introduction

Novel view synthesis (NVS) has played an important role in many everyday applications and is expected to become even more crucial in the future as a foundational technique for augmented reality (AR) and virtual reality (VR). It has also received growing attention in the research community with the introduction of 3D Gaussian Splatting (3DGS) [1] and a series of subsequent developments [2–6]. Although 3DGS achieves real time rendering and high visual quality, its reliance on slow per-scene tuning significantly limits its practical use in real world scenarios.

To address this limitation, feed-forward 3DGS [7, 8] has been introduced to improve the usability of 3DGS. Unlike conventional 3DGS approaches that rely on slow per-scene backward optimization, feed-forward 3DGS introduces an "encoder" to extract scene dependent features from input images, allowing the model to benefit from large scale training and predict 3DGS in a single forward pass. Despite notable progress [9–13], these methods remain constrained to a small number of input views, limiting their ability to fully utilize datasets with dense multiple input views [14–17]. For example, our experiments show that the state-of-the-art model DepthSplat [12] suffers a significant performance drop and increased computational cost as input views become denser (see Tab. 1 and Fig. 1), . While better engineering might alleviate this memory issue to some extent, it cannot address the huge

---

[1]Corresponding authors: wangweijie@zju.edu.cn, donny.chen@outlook.sg.
[2]This work was conducted while D. Y. Chen was affiliated with Monash University.

39th Conference on Neural Information Processing Systems (NeurIPS 2025).

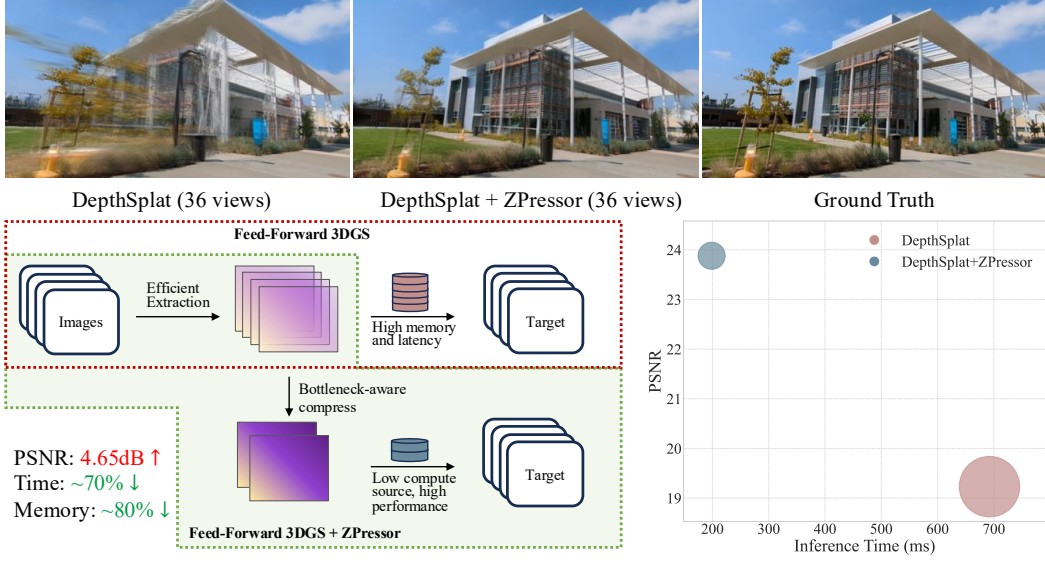

Figure 1: We visualize the result of DepthSplat [12] with 36 view input for novel view synthesis and after adding ZPressor. We report PSNR, inference time, and memory usage before and after simply adding the ZPressor, where the radius of the bubble corresponds to memory.

performance degradation. Upon examining the architecture of several representative feed-forward 3DGS models [7, 8, 12], we identify the *limited capacity of the whole network from image to 3DGS* as the root cause. By design, it struggles to scale to denser inputs due to representation overload from excessive feature tokens and the resulting high computational cost.

Rather than introducing yet another ad-hoc encoder design, this work revisits the feed-forward 3DGS framework with inspiration from the Information Bottleneck (IB) [18] principle. IB offers a theoretical foundation for learning compact representations that preserve only task-relevant information. In the context of NVS with dense input views, we hypothesize that a latent representation can be learned to capture essential scene information while discarding redundant details in dense multi-view inputs. By encouraging the formation of such a compressed yet informative representation, we aim to improve the scalability of feed-forward 3DGS models (detailed in Sec. 3.2). Building on this insight, we propose a lightweight module, termed ZPressor, designed to be seamlessly integrated into the encoder of existing feed-forward 3DGS models to enhance their scalability. Unlike typical efforts that might rely on engineering optimizations such as memory-efficient attention [19] or activation checkpointing [20], our approach adopts a principled perspective grounded in representation learning, aiming to address core architectural limitations under dense input views settings.

To put this idea into practice, ZPressor implements the IB principle by explicitly compressing input view information, as illustrated in Fig. 2. Specifically, we divide the input views into two groups: anchor views and support views. Anchor views serve as the compression states, while information from support views is compressed into them. To ensure the compressed representation retains sufficient scene information, we select anchor views using farthest point sampling to maximize coverage with restricted views. The remaining views are assigned to their nearest anchor based on camera distance, and their features are fused into the anchors through a stack of customized cross-attention blocks. In essence, ZPressor takes multiple input views' features and their corresponding camera poses as input and produces a compact latent representation that preserves scene information. This design is architecture-agnostic and thus can be integrated into various feed-forward 3DGS models.

To validate the effectiveness of ZPressor, we integrate ZPressor into several state-of-the-art feed forward 3DGS models, including pixelSplat [7], MVSplat [8], and DepthSplat [12], and conduct extensive experiments on large-scale benchmarks such as DL3DV-10K [17] and RealEstate-10K [21]. Results show that integrating ZPressor consistently boosts the performance of baseline models under a moderate number of input views (*e.g.*, 12 views), and helps them maintain reasonable accuracy

and computational cost even with very dense inputs (*e.g.*, 36 views, as shown in Fig. 1), where the original models typically degrade dramatically or run out of memory. Our contributions are threefold:

- We provide a fundamental analysis of why existing feed-forward 3DGS models struggle with dense input views, through the lens of the Information Bottleneck principle.
- Inspired by IB, we propose ZPressor, an architecture-agnostic module that can be integrated into the encoder of existing feed-forward 3DGS models to compress input view information.
- Extensive experiments on several large-scale benchmarks show that ZPressor consistently improves the performance of baseline models with a moderate number of input views, and further enhances robustness under dense input settings, where existing models typically degrade significantly.

## 2 Related Work

### 2.1 Information bottleneck and its applications

The challenge of managing and processing vast quantities of information is a central theme in the development of large-scale machine learning models, particularly in the visual domain [22–25]. The Information Bottleneck principle [18] formalizes the problem of extracting a compressed representation $Z$ from the input $X$, such that $Z$ is maximally informative about the target $Y$. The IB principle was subsequently extended to the domain of deep learning [26–29], VIB [30] providing a tractable lower bound on the IB objective, bridging the gap between the theoretical IB principle and practical deep learning applications. A series of works have applied the IB principle to multi-view inputs [31–33], extract information that is common or shared across multiple views while discarding view-specific or redundant information. The drive towards efficient 3D scene reconstruction, especially within the context of 3DGS [1], has also seen the adoption of information-theoretic ideas. StreamGS [34] tackle redundancy in image streams by merging superfluous Gaussians through cross-frame feature aggregation, while other works [35, 36] focus on compressing the learned 3D Gaussians. While demonstrating the effectiveness and necessity of compression in multi-view data and 3D reconstruction, none of the existing works have explored it in the context of feed-forward 3DGS. Our work aims to bridge this gap by introducing the lens of IB for information compression in this area.

### 2.2 Optimization-based NeRF and 3DGS

Traditional novel view synthesis (NVS) methods primarily rely on image blending techniques [37, 38]. More recently, neural network-based approaches [39–42] have advanced NVS by integrating it with deep learning models. In particular, NeRF [42] employs an MLP to map 3D spatial locations and viewing directions to radiance color and volume density. Numerous works [43–48] have sought to improve NeRF's efficiency and reconstruction quality. However, its reliance on volume rendering [49] hinders rendering speed, limiting its practicality in real-world applications. Recently, 3D Gaussian Splatting (3DGS)[1] and its variants[2, 36, 50–52] have emerged as efficient solutions for large-scale scene reconstruction and synthesis, offering explicit representations and fast rasterization-based rendering that outperform NeRF's slower volumetric approach. Nonetheless, its requirement for slow per-scene optimization still poses challenges for deployment in downstream tasks.

### 2.3 Feed-Forward NeRF and 3DGS

To address the limitation of slow per-scene optimization, PixelNeRF [53] pioneered the feed-forward NeRF (*a.k.a.* generalizable NeRF) by introducing an additional network that directly encodes input views into a NeRF representation. This allows the model to benefit from large-scale training and predict a scene representation in a single forward pass. This direction has since seen significant advancements [54–58], offering a promising path toward practical NeRF deployment. This paradigm has recently been extended to real-time rendering with 3DGS [1] replacing NeRF [42]. Among them, pixelSplat [7] pioneered the feed-forward 3DGS approach by combining epipolar transformers with depth prediction to predict 3D Gaussians from two input views. MVSplat [8] proposed an efficient cost-volume-based fusion strategy to improve multi-view reconstruction, while DepthSplat [12] leveraged monocular depth features to better recover fine 3D structures from sparse inputs. Despite the growing number of feed-forward 3DGS models [59–63], most follow a pixel-aligned design, where

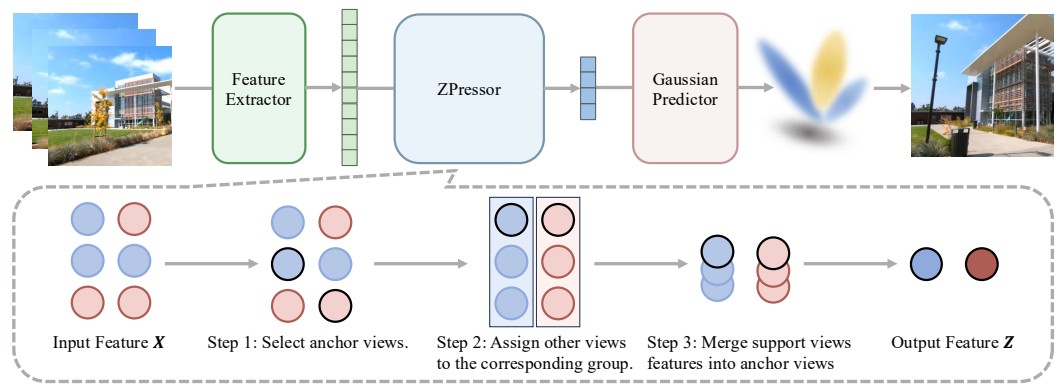

Figure 2: **Overview of ZPressor for Feed-Forward 3DGS.** Our proposed ZPressor is an architecture-agnostic module designed for feed-forward 3DGS frameworks. It addresses the challenge of processing dense input views by strategically grouping input view features $\mathcal{X}$ based on selected anchor views, then features within each respective group are compressed as $\mathcal{Z}$.

the number of 3D Gaussians scales linearly with the number of input views. This leads to significant memory and computational overhead as input views increase. While works like FreeSplat [10] and GGN [11] attempt to reduce the number of Gaussians by merging them via cross-view projection checking, they lack a principled framework. In contrast, our work provides a theoretical perspective on the representation overload problem in feed-forward 3DGS by introducing the IB [18] principle. And we propose an architecture-agnostic module ZPressor that can be seamlessly integrated into existing feed-forward 3DGS models to improve performance under dense input view settings.

## 3 Methodology

### 3.1 Overview of ZPressor

Our ZPressor is an architecture-agnostic module for compressing the multi-view inputs of feed-forward 3DGS, as illustrated in Fig. 2. Formally, given $K$ input views $\mathcal{V} = \{\mathbf{V}_i\}_{i=1}^{K}$ where $\mathbf{V}_i \in \mathbb{R}^{H \times W \times 3}$ and their corresponding camera poses $\mathcal{P} = \{\mathbf{P}_i\}_{i=1}^{K}$, our ZPressor takes the extracted features from each view as input:

$$\mathcal{X} = \{\mathbf{F}_i\}_{i=1}^{K} = \Phi_{\text{image}}(\mathcal{V}, \mathcal{P}), \quad \mathbf{F}_i \in \mathbb{R}^{\frac{H}{p} \times \frac{W}{p} \times C} \tag{1}$$

where $\Phi_{\text{image}}$ is a pretrained image encoder. Then our ZPressor adaptively compresses these heavy multi-view features into compact ones $\mathcal{Z} = \text{ZPressor}(\mathcal{X})$. Subsequently, we directly unprojects these compact latent representations into 3D space using the camera poses $\mathcal{P}$ and a pixel-aligned Gaussian prediction network $\Psi_{\text{pred}}$ is employed to estimate the Gaussian parameters:

$$\mathcal{Y} = \{(\mu_i, \boldsymbol{\Sigma}_i, \alpha_i, \mathbf{c}_i)\}_{i=1}^{H \times W \times K} = \Psi_{\text{pred}}(\mathcal{Z}, \mathcal{P}). \tag{2}$$

The Gaussian parameters $\mathcal{Y}$ include mean $\mu$, opacity $\alpha$, covariance matrix $\boldsymbol{\Sigma}$, and color $\mathbf{c}$, while this pixel-aligned prediction results in a linear increase in Gaussian primitives with more input views, constraining the model's input capacity.

### 3.2 Information Analysis of Feed-Forward 3DGS

Existing feed-forward 3D Gaussian Splatting networks suffer from a dramatic performance drop and an exponential increase in computational cost when the amount of input view information grows (see Tab. 1), primarily due to information redundancy and the lack of adaptive information compression mechanisms. Specifically, the total information of the scene is represented by the joint entropy $H(\mathbf{F}_1, \mathbf{F}_2, ..., \mathbf{F}_K)$, which is not merely the sum of the individual entropies of the features from all views, *i.e.*, $\sum_{i=1}^{K} H(\mathbf{F}_i)$. Therefore, there is a significant amount of redundant information in the

features, and it is crucial to remove the irrelevant information after feature extraction while preserving its predictive power, which allows for the efficient use of information from the input views.

A priciple way for modeling this is the Information Bottleneck (IB) [18], which minimizes the IB scores as:

$$\min_{\mathcal{Z}} IB = \underbrace{\beta \, I(\mathcal{X}, \, \mathcal{Z})}_{\text{Compression Score}} - \underbrace{I(\mathcal{Z}, \, \mathcal{Y})}_{\text{Prediction Score}} , \tag{3}$$

where $\beta \geq 0$ controls the balance between compression and prediction, and $I(\cdot, \cdot)$ is the mutual information. The *Compression Score* component, $\beta I(\mathcal{X}, \mathcal{Z})$, encourages $\mathcal{Z}$ to be a concise representation of the input $\mathcal{X}$. Minimizing $I(\mathcal{X}, \mathcal{Z})$ means reducing the amount of information $\mathcal{Z}$ carries about $\mathcal{X}$, which leads to better compression and enhanced efficiency. The *Prediction Score* component, $I(\mathcal{Z}, \mathcal{Y})$, measures the predictive power of the latent feature $\mathcal{Z}$ with respect to the target variable $\mathcal{Y}$. Maximizing this term ensures that $\mathcal{Z}$ retains sufficient task-relevant information about $\mathcal{Y}$, which is vital for maintaining or improving prediction accuracy.

As shown in Fig. 2, the *Prediction Score* is typically modelled by the Gaussian predictor. In the next section, we introduce ZPressor, a novel module applied concatenated with the image encoder to model the *Compression Score* through three consecutive designs, yielding a compact presentation.

## 3.3 Information Compression Module: ZPressor

To ensure architecture-agnostic integration, compression is performed along the view dimension, rather than being entangled with the design of a specific model. Concretely, given the $K$ encoded views $\mathcal{X}$, we first divide them into $N$ *anchor views* $\mathcal{X}_{\text{anchor}} = \{\mathbf{F}_{a_i}\}_{i=1}^{N}$ and $M$ *support views* $\mathcal{X}_{\text{support}} = \{\mathbf{F}_{s_j}\}_{j=1}^{M}$, where $a_i, s_j \in \{1, \dots, K\}$, $\mathcal{X}_{\text{anchor}} \cap \mathcal{X}_{\text{support}} = \emptyset$, and $\mathcal{X}_{\text{anchor}} \cup \mathcal{X}_{\text{support}} = \mathcal{X}$. Here, $\mathcal{X}_{\text{anchor}}$ are expected to capture the essential information of the scene, while $\mathcal{X}_{\text{support}}$ contain supporting context but may include redundancy. Compression is then achieved by *fusing* the features of support views into their *corresponding* anchor views. This raises three design questions: **1)** how to select the anchors $\mathcal{X}_{\text{anchor}}$, **2)** how to assign $\mathcal{X}_{\text{support}}$ to specific anchors, and **3)** how to fuse the information from $\mathcal{X}_{\text{support}}$ into their designated $\mathcal{X}_{\text{anchor}}$.

**Anchor view selection** (Fig. 2 Step 1). Given a set of camera positions $\mathcal{T} = \{\mathbf{T}_i\}_{i=1}^{K} \in \mathbb{R}^{K \times 3}$ calculated from camera parameters $\mathcal{P}$, where $f \colon \mathbf{T} \to \mathbf{F}$ such that $f$ is a bijective mapping, we first add a random view to the anchor view list $\mathcal{S} = \{\mathbf{T}_{a_1}\}$, where $\mathbf{T}_{a_1} \sim \text{Uniform}(\mathcal{T})$.

Subsequent anchor views are iteratively selected as the view with the greatest distance to the current anchor view set $\mathcal{S}$:

$$\mathbf{T}_{a_{i+1}} = \arg \max_{\mathbf{T}_j \in \mathcal{T} \setminus \mathcal{S}} \left( \min_{\mathbf{T}_k \in \mathcal{S}} d(\mathbf{T}_j, \mathbf{T}_k) \right), \tag{4}$$

where $d(\cdot, \cdot)$ denotes the Euclidean distance. This procedure is repeated until $N$ anchors are selected, resulting in a diverse and representative set $\mathcal{X}_{\text{anchor}}$.

**Support-to-anchor assignment** (Fig. 2 Step 2). Once anchors are selected, each support view is assigned to its nearest anchor based on camera position. This ensures that support views, which capture complementary scene details, are grouped with the most spatially relevant anchor views, thereby ensuring the effectiveness of information fusion. Formally, the cluster assignment to the $i$-th anchor view can be denoted as:

$$\mathcal{C}_i = \{f(\mathbf{T}) \in \mathcal{X}_{\text{support}} \mid \|\mathbf{T} - \mathbf{T}_{a_i}\| \leq \|\mathbf{T} - \mathbf{T}_{a_j}\|, \forall j \neq i\} \tag{5}$$

**Views information fusion** (Fig. 2 Step 3). Upon obtaining the anchor-support view-based clusters, we aim to fuse the information within each cluster. The fusion mechanism should satisfy the following properties: **1)** use the anchor views as the base, while effectively integrating information from the support views to enhance them, and **2)** capture the similarity between the two sets of views, maintaining compactness while avoiding redundancy.

Building on the above analysis, the fusion within each cluster can be effectively achieved via cross-attention:

$$\mathcal{Z} = \text{Attention}(Q, K, V), \quad Q \leftarrow \mathcal{X}_{\text{anchor}}, \quad K, V \leftarrow \mathcal{X}_{\text{support}}, \tag{6}$$

where the features $\mathcal{X}_{\text{anchor}}$ extracted from the anchor views are used as queries, while the features $\mathcal{X}_{\text{support}}$ from the support views provide keys and values. In this way, information from the support views is effectively integrated into the anchor views, satisfying the first property. Additionally, it ensures gradient flow from the prediction to both anchor and support views, enabling the model to capture correlations between the two sets of views, thus satisfying the second property. Moreover, since most existing feed-forward 3DGS encoders adopt Transformer-based architectures, the attention module can be readily integrated into these frameworks.

**Training.** Upon obtaining the compressed information $\mathcal{Z}$, we can apply the IB principle to regularize the information flow. Recall that, as in Eq. (3), our current goal should be to minimize the *Compression Score* $I(\mathcal{X}_{\text{anchor}}, \mathcal{X}_{\text{support}}; \mathcal{Z})$ and maximize *Prediction Score* $I(\mathcal{Z}; \mathcal{Y})$. For the *Compression Score*, we apply a constraint on its complexity, *i.e.*, setting the number of anchor views $N$ to a value acceptable for training. For the *Prediction Score*, we have

$$I(\mathcal{Z}; \mathcal{Y}) = H(\mathcal{Y}) - H(\mathcal{Y} \mid \mathcal{Z}), \tag{7}$$

where $H(\mathcal{Y})$ is a constant representing the 3D Gaussians that model the underlying 3D scene. Hence, maximizing $I(\mathcal{Z}; \mathcal{Y})$ is equivalent to minimizing $H(\mathcal{Y} \mid \mathcal{Z})$, which essentially encourages the predicted 3D Gaussians $\mathcal{Y}$ to resemble the original scene.

Then we incorporate the IB principle into feed-forward 3DGS training, using the efficient estimate of Eq. (3) following [26, 30]. Formally, the training objective is defined as:

$$\mathcal{L} = \mathop{\mathbb{E}}_{\mathcal{Z} \sim p_\theta(\mathcal{Z}|\mathcal{X})} \big[ -\log q_\phi(\mathcal{Y} \mid \mathcal{Z}) \big] + \beta \mathop{\mathbb{E}}_{\mathcal{X}} \big[ \text{KL}\big[ p_\theta(\mathcal{Z} \mid \mathcal{X}), \, r(\mathcal{Z}) \big] \big] \tag{8}$$

where $\phi$ denotes the parameters of 3D Gaussians prediction network and $-\log q_\phi(\mathcal{Y} \mid \mathcal{Z})$ can be modeled by the rendering loss (such as the MSE and LPIPS loss), $\theta$ denotes the parameters of the network preceding $\mathcal{Z}$ and $p_\theta(\mathcal{Z} \mid \mathcal{X})$ is the posterior probability estimate of $\mathcal{Z}$, $r(\mathcal{Z}) \sim \mathcal{N}(\mathcal{Z} \mid \mu_0, \Sigma_0)$ is the Gaussian prior of $\mathcal{Z}$.

In our implementation, we append an additional self-attention layer to further enhance information flow within each cluster. In addition, we stack several blocks (each containing both cross- and self-attention) to further improve the effectiveness of the IB principle. These two general-purpose engineering techniques help boost performance, as shown in Tab. 4.

## 4 Experiments

### 4.1 Experimental Settings

**Datasets.** We validate the effectiveness of our ZPressor on the NVS task, following existing works [7, 8, 12], and conduct experiments on two large-scale datasets: DL3DV-10K (DL3DV) [17] and RealEstate10K (RE10K) [21]. DL3DV is a challenging large-scale dataset that contains 51.3 million frames from 10,510 real scenes. We used 140 benchmark scenes for testing and the remaining 9896 scenes for training, with filtering applied to ensure that there is strictly no overlap between the training and test sets. RE10K offers a large-scale collection of indoor home tour clips, comprising 10 million frames from around 80,000 video clips sourced from public YouTube videos. It is split into 67,477 training and 7,289 testing scenes. Both datasets feature real-world captured scenes, with camera intrinsics and extrinsics reconstructed using COLMAP [64, 65].

**Baselines and metrics.** To evaluate the effectiveness and flexibility of our proposed ZPressor, we integrate it as a module into three representative baselines, including DepthSplat [12], MVSplat [8], and pixelSplat [7]. For fair comparisons, we insert ZPressor into the official implementations of each baseline and strictly follow the experimental settings described in their respective papers. Specifically, we compare with DepthSplat on DL3DV, following its training strategy by first pre-training on RE10K

Table 1: **Quantitative comparisons on DL3DV [17].** We evaluate both DepthSplat [12] and DepthSplat with ZPressor with 12, 16, 24, 36 input views and test on eight target novel views.

| Views | Methods | PSNR↑ | SSIM↑ | LPIPS↓ |
|---|---|---|---|---|
| 36 views | DepthSplat | 19.23 | 0.666 | 0.286 |
| | DepthSplat + ZPressor | **23.88**+4.65 | **0.815**+0.149 | **0.150**-0.136 |
| 24 views | DepthSplat | 20.38 | 0.711 | 0.253 |
| | DepthSplat + ZPressor | **24.26**+3.88 | **0.820**+0.109 | **0.147**-0.106 |
| 16 views | DepthSplat | 22.07 | 0.773 | 0.195 |
| | DepthSplat + ZPressor | **24.25**+2.18 | **0.819**+0.046 | **0.147**-0.047 |
| 12 views | DepthSplat | 23.32 | 0.807 | 0.162 |
| | DepthSplat + ZPressor | **24.30**+0.97 | **0.821**+0.014 | **0.146**-0.017 |

Table 2: **Quantitative comparisons on RE10K [21].** We test pixelSplat [7] and MVSplat [8] on eight target views, "OOM" represent that model cannot infer on an 80G GPU.

| Views | Methods | PSNR↑ | SSIM↑ | LPIPS↓ |
|---|---|---|---|---|
| 36 views | pixelSplat | OOM | OOM | OOM |
| | pixelSplat + ZPressor | **26.59** | **0.849** | **0.225** |
| | MVSplat | 24.19 | 0.851 | 0.155 |
| | MVSplat + ZPressor | **27.34**+3.15 | **0.893**+0.042 | **0.113**-0.042 |
| 24 views | pixelSplat | OOM | OOM | OOM |
| | pixelSplat + ZPressor | **26.72** | **0.851** | **0.223** |
| | MVSplat | 25.00 | 0.871 | 0.137 |
| | MVSplat + ZPressor | **27.49**+2.49 | **0.895**+0.024 | **0.111**-0.026 |
| 16 views | pixelSplat | OOM | OOM | OOM |
| | pixelSplat + ZPressor | **26.81** | **0.853** | **0.221** |
| | MVSplat | 25.86 | 0.888 | 0.120 |
| | MVSplat + ZPressor | **27.60**+1.74 | **0.896**+0.008 | **0.110**-0.010 |
| 8 views | pixelSplat | 26.19 | 0.852 | **0.215** |
| | pixelSplat + ZPressor | **26.86**+0.67 | **0.854**+0.002 | 0.219+0.004 |
| | MVSplat | 26.94 | **0.902** | **0.107** |
| | MVSplat + ZPressor | **27.72**+0.78 | 0.897-0.005 | 0.109+0.002 |

and then fine-tuning on DL3DV. Comparisons with MVSplat and pixelSplat are conducted on RE10K. Following these baselines, we report quantitative results using PSNR, SSIM [66], and LPIPS [67]. As our module focuses on information compression, we additionally report model efficiency in terms of runtime and memory consumption.

**Implementation details.** We use the same computing resources to train the baseline and our method. Due to the memory limit, we use 6 context views for DepthSplat and MVSplat, and 4 context views for pixelSplat. For all of our experiments, we adopted the same learning rate as the baseline, utilized the AdamW optimizer, and trained the models for 100,000 steps on A800 GPUs. Following the setting of the baseline, we use the $256 \times 256$ input resolution on RE10K, and $256 \times 448$ input resolution on DL3DV. All training losses match those of the baseline, with no additional data or regularization introduced.

## 4.2 SoTA Comparisons and IB Analysis

**Comparisons with SoTA models.** We train all models on DL3DV and RE10K using 12 input views with 6 anchor views set to our ZPressor, and evaluate them under varying numbers of input views ranging from 8 to 36. As shown in Tab. 1 and Tab. 2, integrating ZPressor into DepthSplat, MVSplat, and pixelSplat consistently improves their performance across all input view settings and evaluation metrics, demonstrating the effectiveness of our approach.

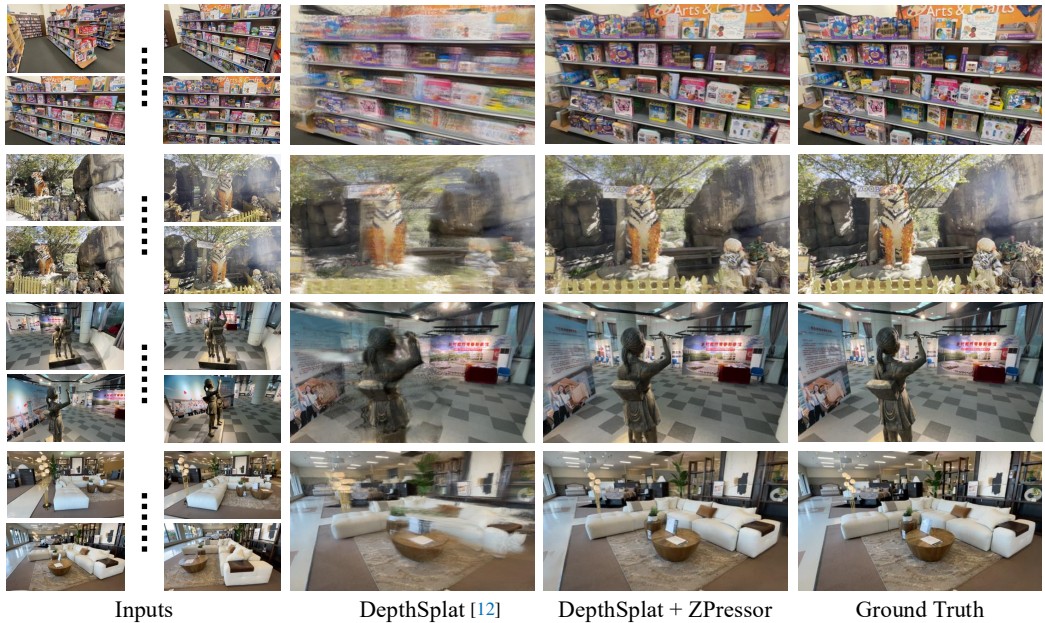

| Inputs | DepthSplat [12] | DepthSplat + ZPressor | Ground Truth |

Figure 3: **Qualitative comparison on DL3DV [17] under dense input conditions (36 views).** DepthSplat [12] performs poorly due to redundancy in dense views, ZPressor effectively compresses this information, achieving significantly improved visual results.

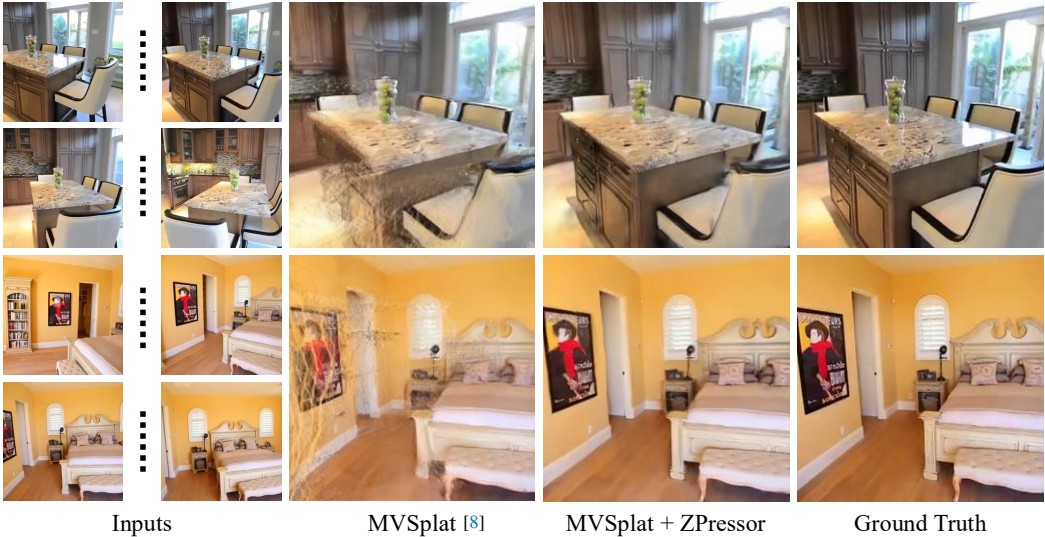

| Inputs | MVSplat [8] | MVSplat + ZPressor | Ground Truth |

Figure 4: **Qualitative comparison on RE10K [21] with 36 input views.** MVSplat [8] with ZPressor performs the best in all cases.

Notably, the performance gain becomes more significant as the number of input views increases. This is because existing feed-forward 3DGS models struggle with dense inputs due to representation overload, leading to performance degradation. In contrast, ZPressor mitigates this issue by compressing the input through redundancy suppression while preserving essential information, improving model robustness, and maintaining strong performance under dense input settings. Moreover, we observe that pixelSplat fails to run with more than 8 input views due to out-of-memory (OOM) caused by the large number of predicted pixel-aligned 3D Gaussians. In contrast, our ZPressor helps merge input information, reducing the number of predicted Gaussians and enabling testing with up to 36

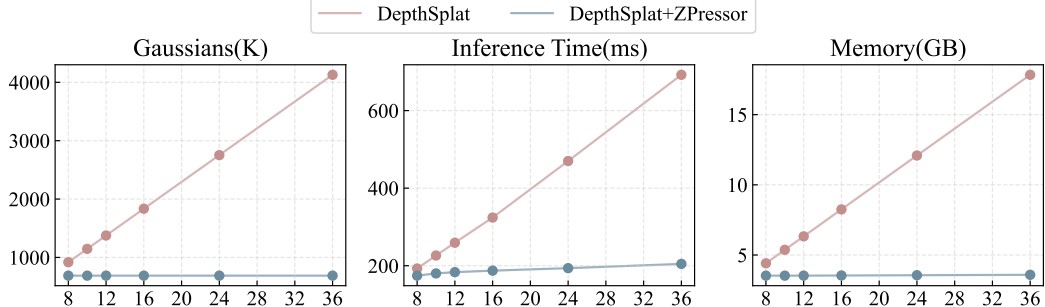

Figure 5: **Efficiency analysis.** We report the number of Gaussians (K), inference time (ms) and peak memory (GB) of DepthSplat [12] and DepthSplat with ZPressor.

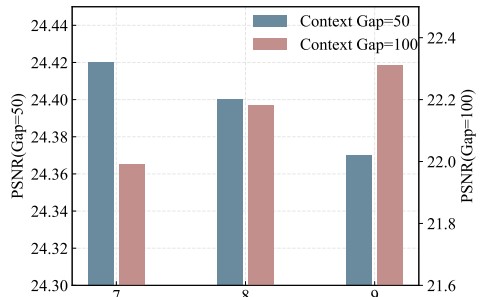

Figure 6: **Analysis of the bottleneck constraint.** We compare the performance of ZPressor in different scale of scene coverage.

Table 3: **Analysis of Information Fusion.** "default" denotes our setting where support views are fused into anchor views. "w/o fusion" removes the fusion step, and "fuse anchors" fuses repeated anchor views instead. "default" performs best, indicating that ZPressor improves performance by effectively fusing complementary information from the support views.

| Method | PSNR↑ | SSIM↑ | LPIPS↓ |
|---|---|---|---|
| default | **24.30** | **0.821** | **0.146** |
| fuse anchors | 24.23 | 0.817 | 0.148 |
| w/o fusion | 23.80 | 0.810 | 0.162 |

views. These observations are further supported by the qualitative comparisons shown in Fig. 3 and Fig. 4, where DepthSplat and MVSplat exhibit noticeable artifacts under 36 input views, whereas their ZPressor-augmented versions produce significantly cleaner renderings.

**Analysis of model efficiency.** Compressing the input view information not only improves robustness and performance, but also enhances efficiency. To validate this, we compare the model efficiency of DepthSplat with and without ZPressor under 480P resolution, evaluating the number of 3D Gaussians, the test-time inference latency, and the peak memory usage. As shown in Fig. 5, the benefits of integrating ZPressor are clear. In particular, as the number of context views increases, the baseline model's predicted 3D Gaussian numbers, memory usage, and inference time all grow linearly, whereas ZPressor helps maintain stable resource consumption across all aspects.

**Analysis of the bottleneck constraint.** In the context of NVS, the information corresponds to the overall region that the scene covers. Since we evaluated on the static video data DL3DV, longer sequences usually cover larger regions. Therefore, we use the frame distance between input views as a proxy for both scene coverage and information content.

We can then analysis the effect of bottleneck constraint (the number of anchor views) under scenes with varying information content by adjusting the frame distance between input views. As shown in Fig. 6, we conduct experiments under two settings, Context Gap 50 (CG50, in blue) and Context Gap 100 (CG100, in pink), where context gap refers to the frame distance between input views. When the scene's information content is relatively low (e.g., small camera baseline, similar views, as proxied by a small CG like 50), a smaller number of anchor views (e.g., 7) is already sufficient to capture the essential scene information. Adding more anchor views in such a scenario might introduce redundancy or ambiguity, as these additional anchors might not observe genuinely new regions but rather re-observe already covered areas from slightly different perspectives. This could lead to a less compact and potentially noisier latent representation, hence the performance drop. In contrast, for scenes with higher information content (larger CG like 100), more anchor views are beneficial as they help cover more diverse perspectives and capture richer scene details. These results

Table 4: **Ablation study of our method with DepthSplat [12] on the DL3DV dataset [17].** Models are evaluated by rendering eight novel views using 12 input views.

| Methods | PSNR↑ | SSIM↑ | LPIPS↓ | Time (s) | Peak Memory (GB) |
|---|---|---|---|---|---|
| DepthSplat + ZPressor | **24.30** | **0.821** | **0.146** | 0.184 | 3.80 |
| w/o multi-blocks | 24.18 | 0.817 | 0.149 | **0.140** | **3.79** |
| w/o self-attention | 23.85 | 0.810 | 0.156 | 0.183 | 3.80 |
| DepthSplat | 23.32 | 0.808 | 0.162 | 0.260 | 6.80 |

highlight the effectiveness of our ZPressor as an instantiation of the IB principle and show that the information bottleneck is critical in balancing compression and information preservation.

**Analysis of information fusion.** To further confirm that our ZPressor effectively fuses information from support views into anchor views rather than simply discarding it, we conduct experiments by varying the fusion strategy. As shown in Tab. 3, removing the information fusion step ("w/o fusion") leads to a performance drop, highlighting the importance of fusing support views into anchor views. To ensure that the introduction of support information does result in a performance improvement, we conduct a control experiment by fusing repeated anchor views instead of support views ("fuse anchors"). Since this does not introduce new information, its performance is lower than our default setting, which achieves the best results by fusing complementary information. These comparisons further validate that our design effectively implements the IB principle.

### 4.3 Ablation Study

As mentioned at the end of Sec. 3, the default ZPressor uses several stacked attention blocks, each combining self-attention and cross-attention. This section presents an ablation study to validate our design.

We report results on DL3DV with 12 input views, following the setting in Tab. 1, using DepthSplat as the backbone. As shown in Tab. 4, removing the stacking design and using only one block ("w/o multi-blocks") slightly degrades performance, suggesting that stacking improves fusion of information from support to anchor views. Additionally, removing the self-attention ("w/o self attention") also reduce performance, showing that self-attention complements cross-attention by enhancing internal feature interactions. Overall, all variants of ZPressor outperform the DepthSplat baseline, confirming the existence of an information bottleneck in feed-forward 3DGS models and the effectiveness of our ZPressor in addressing it.

## 5 Conclusion

We have provided a fundamental analysis of the model capacity limitations in existing feed-forward 3DGS models through the lens of the Information Bottleneck principle. Building on this insight, we have introduced ZPressor, a lightweight, architecture-agnostic module that efficiently compresses multi-view inputs, enabling models to overcome inherent limitations and scale to handle more input views. We have validated our ZPressor by integrating it into several representative feed-forward 3DGS models. Our experiments on several large-scale benchmarks have demonstrated that ZPressor not only consistently improves the performance of existing models under moderate view settings, but also helps them maintain competitive efficiency under denser inputs. We believe that ZPressor significantly enhances the scalability and practicality of feed-forward 3DGS models, opening the door to more effective applications in real-world scenarios.

**Limitation and discussion.** Our ZPressor may be less effective in extremely dense view settings. For example, given 1000 input views, ZPressor can only compress them to around 50 views in order to maintain the information compactness as regularized by the IB principle. However, handling 50 views of 3D Gaussians still presents significant computational challenges for typical GPUs. Future work could explore combining ZPressor with 3D Gaussian merging or memory-efficient rendering to extend feed-forward 3DGS to handle extremely dense input views.

**Acknowledgements.** We thank Biao Wu for his enlightening discussions and inspiration on network architecture.

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

# A  More Experimental Analysis

## A.1  Cross Dataset Generalization

Following MVSplat [8], we conducted experiments using a pretrained model on the RealEstate10K (RE10K) dataset [21] (as detailed in Tab. 2) and tested its performance on the ACID dataset [68] to evaluate the generalization capabilities of our proposed ZPressor across diverse datasets. As demonstrated in Table A, MVSplat with ZPressor exhibits remarkable efficacy in cross-dataset generalization. Notably, this performance advantage becomes progressively more pronounced with an increasing number of input views.

Table A: **Quantitative comparison on ACID [68] with trained model on RE10K.** Trained on indoor scenes (RE10K), MVSplat [8] and pixelSplat [7] with ZPressor perform much better as evaluated on the ACID dataset.

| Views | Methods | PSNR↑ | SSIM↑ | LPIPS↓ |
|---|---|---|---|---|
| 36 views | pixelSplat | OOM | OOM | OOM |
| | pixelSplat + Ours | **27.78** | **0.823** | **0.238** |
| | MVSplat | 24.89 | 0.812 | 0.179 |
| | MVSplat + Ours | **28.16**$_{+3.27}$ | **0.853**$_{+0.041}$ | **0.145**$_{-0.034}$ |
| 24 views | pixelSplat | OOM | OOM | OOM |
| | pixelSplat + Ours | **27.91** | **0.825** | **0.235** |
| | MVSplat | 25.46 | 0.829 | 0.167 |
| | MVSplat + Ours | **28.33**$_{+2.87}$ | **0.856**$_{+0.027}$ | **0.142**$_{-0.025}$ |
| 16 views | pixelSplat | OOM | OOM | OOM |
| | pixelSplat + Ours | **27.97** | **0.826** | **0.234** |
| | MVSplat | 26.08 | 0.844 | 0.156 |
| | MVSplat + Ours | **28.42**$_{+2.34}$ | **0.858**$_{+0.014}$ | **0.141**$_{-0.015}$ |
| 8 views | pixelSplat | 26.69 | 0.807 | 0.260 |
| | pixelSplat + Ours | **28.05**$_{+1.36}$ | **0.828**$_{+0.021}$ | **0.234**$_{-0.026}$ |
| | MVSplat | 27.89 | **0.864** | **0.140** |
| | MVSplat + Ours | **28.60**$_{+0.71}$ | 0.860$_{-0.004}$ | **0.140**$_{-0.000}$ |

## A.2  Other Selection Strategies

In addition to our final choice of FPS-based selection strategy (Fig. 2 Step 1), we also experimented with other anchor views selection strategies. Below are detailed implementation descriptions for each strategy and a comparison of all strategies' performance in Tab. B.

**Overlap-based selection.** We try overlap-based selection by computing a pairwise overlap matrix between camera views. The overlap is obtained by projecting dense pixel rays from one view into another and averaging the proportion of pixels that fall within the target image plane. The final overlap score between two views is the minimum of the bidirectional projections. We then apply a greedy vectorized algorithm that iteratively selects views with the lowest average overlap to the already chosen set, ensuring maximal scene coverage with minimal redundancy. This procedure produces both the anchor indices and the cluster assignments for the remaining views.

**Pose-free settings.** In the absence of camera poses, we cluster learned per-view tokens instead of 3D positions. We use K-Means in the feature space of global view embeddings. Each token is assigned to its nearest cluster center, and the closest token to the mean of each cluster is selected as the anchor. This pose-free clustering provides a practical alternative for unposed datasets, as the anchor set is derived entirely from image content rather than camera geometry.

**K-Means-based selection.** K-Means-based selection groups view positions directly. Given the 3D camera centers of all input views, we apply K-Means with a fixed number of groups and identify the anchor of each cluster as the view closest to its centroid. Each remaining view is assigned

Table B: **Quantitative comparison of different selection strategies integrated with Depth-Splat [12].** All variants are evaluated with 36 test views on DL3DV [17].

| Methods | PSNR↑ | SSIM↑ | LPIPS↓ |
|---|---|---|---|
| DepthSplat | 19.23 | 0.666 | 0.286 |
| DepthSplat + Ours (overlap-based) | 21.49$_{+2.26}$ | 0.727$_{+0.061}$ | 0.194$_{-0.092}$ |
| DepthSplat + Ours (pose-free) | 22.81$_{+3.58}$ | 0.791$_{+0.125}$ | 0.174$_{-0.112}$ |
| DepthSplat + Ours (K-Means-based) | 22.84$_{+3.61}$ | 0.789$_{+0.123}$ | 0.175$_{-0.111}$ |
| DepthSplat + Ours | **23.88**$_{+4.65}$ | **0.815**$_{+0.149}$ | **0.150**$_{-0.136}$ |

to its nearest centroid, producing compact and spatially coherent groups. This procedure is fully deterministic given the random seed and provides stable anchors that reflect the geometric distribution of input cameras.

## A.3 Comparison with Confidence-based Pruning

We compare ZPressor against confidence-based pruning after predicting Gaussians in MVSplat [8]. Confidence pruning removes input views based on their predicted reliability scores, with a fixed pruning ratio controlling the proportion of views retained. When maintaining the same number of Gaussians, the aggressive removal of views leads to a substantial performance drop due to insufficient multi-view support. At a pruning ratio of $0.5$, the method achieve the best accuracy, but the gain remains limited because the retained subset does not guarantee spatial coverage of the scene. In contrast, ZPressor consistently compresses all input views into compact latent anchors while maintaining balanced coverage, which yields superior accuracy without discarding information.

Table C: **Comparison with confidence-based pruning (CP) in MVSplat [8] under 24 views on RE10K.** For a fair comparison, the prune ratio for CP is set to $0.75$ to yield a same number of Gaussians as ZPressor, and we also manually adjust the prune ratio to $0.5$ to achieve the best performance. All models are re-trained.

| Methods | PSNR↑ | SSIM↑ | LPIPS↓ |
|---|---|---|---|
| MVSplat | 25.00 | 0.871 | 0.137 |
| MVSplat + CP ($prune\_ratio = 0.75$) | 21.15$_{-3.85}$ | 0.816$_{-0.055}$ | 0.190$_{+0.053}$ |
| MVSplat + CP ($prune\_ratio = 0.5$) | 26.94$_{+1.94}$ | 0.886$_{+0.015}$ | 0.130$_{-0.007}$ |
| MVSplat + Ours | **27.49**$_{+2.49}$ | **0.895**$_{+0.024}$ | **0.111**$_{-0.026}$ |

## A.4 Ablation Study of IB-Loss

We conduct an ablation study to evaluate the role of the compression term derived from the Information Bottleneck formulation. When the compression term is removed ($\beta = 0$), ZPressor still improves significantly over the baseline DepthSplat, confirming that the architectural design alone provides strong benefits. Introducing a small but non-zero coefficient ($\beta = 10^{-5}$) further encourages compact latent representations and yields the best balance between distortion and fidelity. This demonstrates that the IB-inspired loss can serve as a lightweight regularizer to assist compression module learning in achieving better compression results.

Table D: **Ablation of the IB-inspired compression loss on DepthSplat [12] with 36 views on DL3DV [17].** The compression term provides additional regularization that improves overall performance.

| Methods | PSNR↑ | SSIM↑ | LPIPS↓ |
|---|---|---|---|
| DepthSplat | 19.23 | 0.666 | 0.286 |
| DepthSplat + Ours ($\beta = 0$) | 23.43$_{+4.20}$ | 0.806$_{+0.140}$ | 0.165$_{-0.121}$ |
| DepthSplat + Ours ($\beta = 10^{-5}$) | **23.88**$_{+4.65}$ | **0.815**$_{+0.149}$ | **0.150**$_{-0.136}$ |

# B More Implementation Details

**Network architectures.** In Algorithm 1, we provide a detailed description of how ZPressor is integrated into existing feed-forward 3D Gaussian Splatting (3DGS) frameworks [7, 8, 12]. Initially, we select anchor views and their corresponding support views following Algorithm 2 and Eq. (5). The features associated with these views are then processed by an attention-based network. This network is composed of 6 structurally identical blocks, wherein each block encompasses a cross-attention layer, a self-attention layer, and an MLP layer. The cross-attention mechanism operates by employing the anchor features as query, while the support features provide the key and value. Subsequent to this fusion, the resulting features are further refined by the self-attention and MLP layers.

---

**Algorithm 1** Overview of Feed-Forward 3DGS framework with ZPressor.

---

**Input:** $K$ input views $\mathcal{V} = \{V_i\}_{i=1}^{K}$, camera poses $\mathcal{P} = \{P_i\}_{i=1}^{K}$, the number of anchor views $N$, the number of network blocks $h$.
**Output:** Gaussian parameters $\mathcal{Y} = \{(\mu, \Sigma, \alpha, c)\}$.
    $\mathcal{X} \leftarrow \Phi_{image}(\mathcal{V}, \mathcal{P})$
    $\mathcal{X}_{\text{anchor}}, \mathcal{X}_{\text{support}} \leftarrow \mathcal{X}$, with Anchor view selection.
    Assign support views to anchor cluster $\mathcal{C} \leftarrow \mathcal{X}_{\text{support}}$
    Initialize state $\mathcal{Z} \leftarrow \mathcal{X}_{\text{anchor}}$
    **for** $i \leftarrow 1$ to $h$ **do**
        $\mathcal{Z} \leftarrow \text{Attention}(Q, K, V)$, where $Q \leftarrow \mathcal{Z}$   $K, V \leftarrow \mathcal{X}_{\text{support}}$
        $\mathcal{Z} \leftarrow \text{Attention}(Q, K, V)$, where $Q, K, V \leftarrow \mathcal{Z}$
        $\mathcal{Z} \leftarrow \text{MLP}(\mathcal{Z})$
    **end for**
    $\{(\mu_i, \Sigma_i, \alpha_i, \mathbf{c}_i)\} \leftarrow \Psi_{\text{pred}}(\mathcal{Z}, \mathcal{P})$
    **return** $\mathcal{Y} \leftarrow \{(\mu_i, \Sigma_i, \alpha_i, \mathbf{c}_i)\}$

---

**Algorithm 2** Farthest Point Sampling for Anchor View Selection

---

**Input:** Set of view camera positions $\mathcal{T} = \{\mathbf{T}_1, \mathbf{T}_2, ..., \mathbf{T}_K\}$, Number of anchor views $N$
**Output:** Indices of the selected anchor views $\mathcal{S} = \{\mathbf{T}_{a_1}, \mathbf{T}_{a_2}, ..., \mathbf{T}_{a_n}\}$
    Initialize the set of anchor view indices $\mathcal{S} \leftarrow \emptyset$
    Randomly select a random anchor view $\mathbf{T}_{a_1} \in \mathcal{T}$, where $\mathbf{T}_{a_1} \sim \text{Uniform}(\mathcal{T})$
    Add $\mathbf{T}_{a_1}$ to $\mathcal{S}$: $\mathcal{S} \leftarrow \{\mathbf{T}_{a_1}\}$
    **for** $j \leftarrow 2$ to $N$ **do**
        Initialize a dictionary to store minimum distances $D \leftarrow \{\}$
        **for** $k \leftarrow 1$ to $K$ **do**
            **if** $k \notin \mathcal{S}$ **then**
                Calculate the minimum distance $d_k \leftarrow \min_{i \in \mathcal{S}} \|\mathbf{T}_k - \mathbf{T}_i\|_2$
                Store the distance: $D[k] \leftarrow d_k$
            **end if**
        **end for**
        Find the view position $T_{a_j}$ with the maximum minimum distance: $T_{a_j} \leftarrow \arg\max_{k \notin \mathcal{S}} D[k]$
        Add $a_j$ to $\mathcal{S}$: $\mathcal{S} \leftarrow \mathcal{S} \cup \{T_{a_j}\}$
    **end for**
    **return** $\mathcal{S}$

---

To ensure training stability, deviating from traditional Transformer architectures, we employ Pre-Layer Normalization [69] (Pre-LN), which enhances the robustness of the model. Furthermore, system-level advancements have been incorporated to accelerate computation. For example, we employ FlashAttention [19, 70], which uses highly optimized GPU kernels and leverages hardware topology to compute attention in a time- and memory-efficient manner.

**More training details.** We use the first model version of DepthSplat [12] from its October 2024 release. Experimental results obtained with this specific version may exhibit slight variations when compared to the current version. ZPressor was incorporated subsequent to the monocular feature extraction performed by CNN.

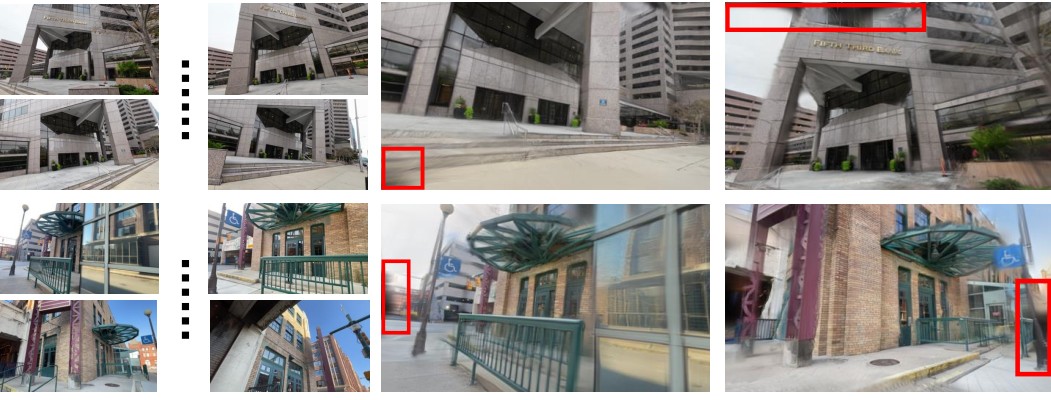

Inputs (~500 views)                         DepthSplat + ZPressor

Figure A: **Limitations.** Visual results from extremely dense input views show slightly poor presentation effect.

Adhering to its original configuration, experiments were conducted at a resolution of $256 \times 448$. The model was initially trained on the RE10K [21] for 100,000 steps and subsequently fine-tuned on the DL3DV [17] for an additional 100,000 steps. We employed the AdamW optimizer [71] with a learning rate of $2 \times 10^{-4}$. The total training duration was approximately two days, and the integration of ZPressor did not significantly alter the original training time of DepthSplat.

Similarly, for MVSplat [8] and pixelSplat [7], ZPressor was integrated after the monocular feature extraction stage. MVSplat utilizes a CNN for feature extraction, whereas pixelSplat employs DINO [24, 25]; this architectural choice in pixelSplat contributes to a marginally higher VRAM consumption compared to the other two baselines. We maintained the model parameter settings as published in their respective original works, training models on the RE10K [21] at a resolution of $256 \times 256$. The learning rate was set to $2 \times 10^{-4}$ for MVSplat and $1.5 \times 10^{-4}$ for pixelSplat, where both of which were trained for 100,000 steps. Notably, due to memory constraints, we trained the pixelSplat model incorporating ZPressor using 4 anchor views, in contrast to the 6 anchor views configured for DepthSplat and MVSplat. The training times for MVSplat and pixelSplat, when augmented with ZPressor, remained comparable to their original durations.

We will **open-source** the complete codebase for ZPressor, our ZPressor-integrated versions of DepthSplat, MVSplat, and pixelSplat, and all associated model checkpoints.

## C   Limitation and Societal Impacts

**Limitation analysis.** As discussed in Sec. 5, ZPressor exhibits limitations when processing scenarios with an extremely high density of input views. Specifically, its efficacy in compressing the information from such dense views through a limited set of anchor views is diminished. To illustrate this, we conducted an experiment on DepthSplat [12] integrated with ZPressor, using approximately 500 images as input. As depicted in Fig. A, the quality of the rendered novel views was perceptibly affected, which can be attributed to an insufficient number of Gaussian primitives to adequately represent the scene under these dense input conditions.

**Potential and negative societal impacts.** ZPressor can significantly reduce the training costs associated with feed-forward 3DGS networks. It enables the processing of a larger number of input views within the same VRAM budget and training duration, delivering high-fidelity rendering results and thereby decreasing energy consumption during the model training process. While the capability to render higher-quality novel views from more densely sampled perspectives positions ZPressor as a valuable tool for augmented reality applications, it is important to acknowledge that the fidelity of the rendering can be compromised by the emergence of artifacts, particularly when processing input views of extremely high density. Consequently, in safety-critical applications, such as the training of

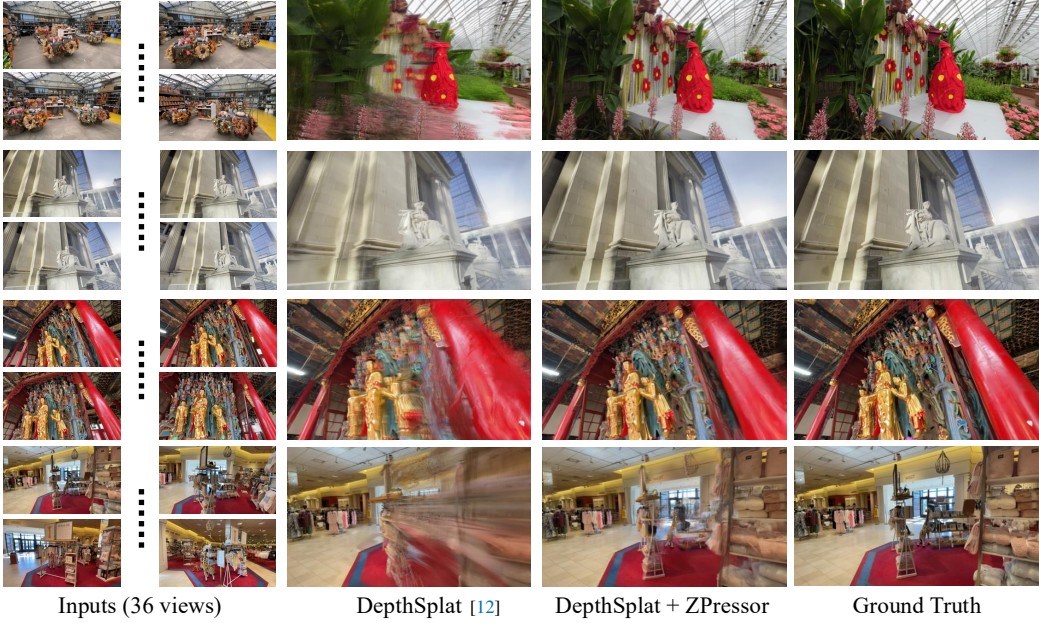

| Inputs (36 views) | DepthSplat [12] | DepthSplat + ZPressor | Ground Truth |

Figure B: **More qualitative comparisons on DL3DV [17] with DepthSplat [12] under 36 input views.** Models with ZPressor performs the best in all cases.

autonomous driving models, the deployment of ZPressor would necessitate the implementation of additional precautionary measures to mitigate potential risks arising from such limitations.

# D    More Visual Comparisons

This section provides additional qualitative comparison results. We present further visualizations for DepthSplat [12] on the DL3DV [17] and MVSplat [8] on the RE10K [21] in Fig. B and Fig. F, with our ZPressor.

Furthermore, to illustrate how ZPressor performs with dense input views, we showcase comparative results. For DepthSplat [12], comparisons between the original framework and DepthSplat augmented with ZPressor are presented for scenarios with 24, 16, and 12 input views in Fig. C, Fig. D, and Fig. E. Similarly, for MVSplat [8], visual comparisons between the original framework and MVSplat integrated with ZPressor are displayed for inputs of 24, 16, and 8 views in Fig. G, Fig. H, and Fig. I. The corresponding quantitative results for these multi-view experiments can be found in Tab. 2.

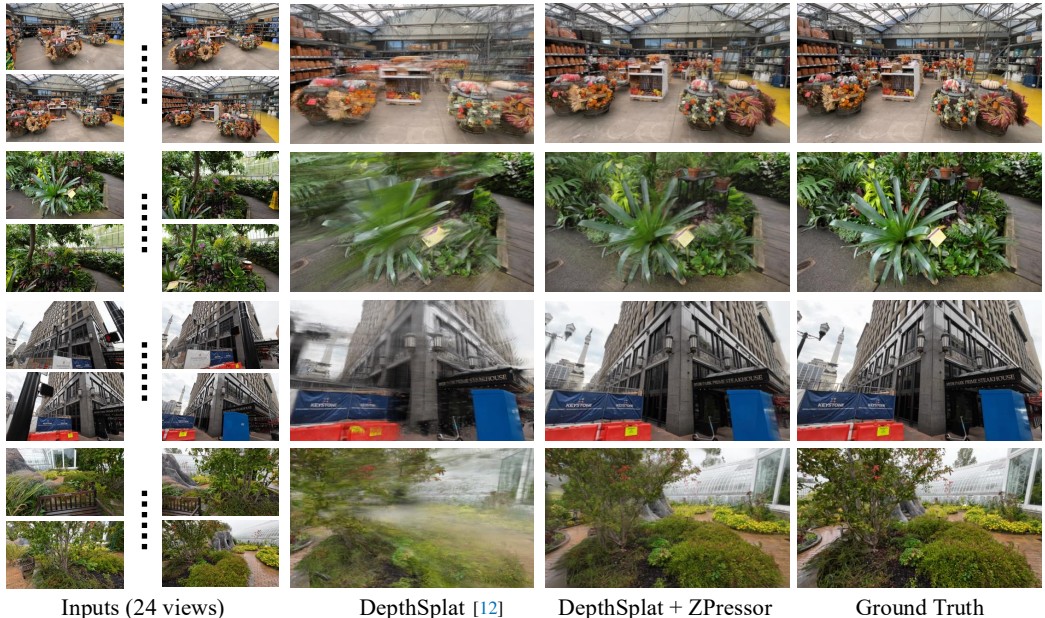

| Inputs (24 views) | DepthSplat [12] | DepthSplat + ZPressor | Ground Truth |

Figure C: **More qualitative comparisons on DL3DV [17] with DepthSplat [12] under 24 input views.**

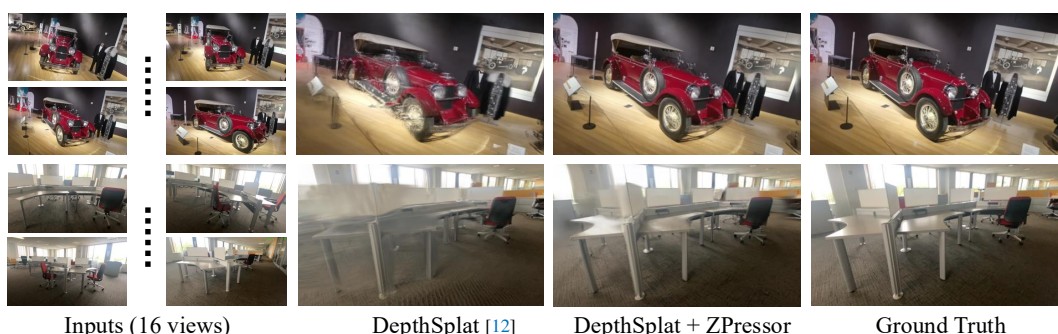

| Inputs (16 views) | DepthSplat [12] | DepthSplat + ZPressor | Ground Truth |

Figure D: **More qualitative comparisons on DL3DV [17] with DepthSplat [12] under 16 input views.**

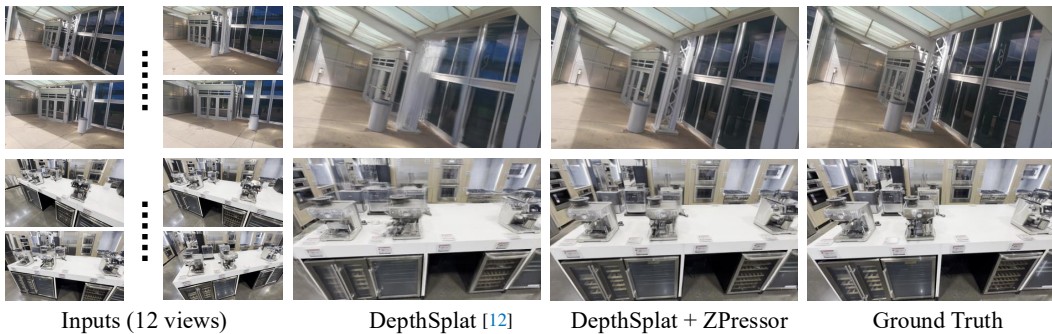

| Inputs (12 views) | DepthSplat [12] | DepthSplat + ZPressor | Ground Truth |

Figure E: **More qualitative comparisons on DL3DV [17] with DepthSplat [12] under 12 input views.**

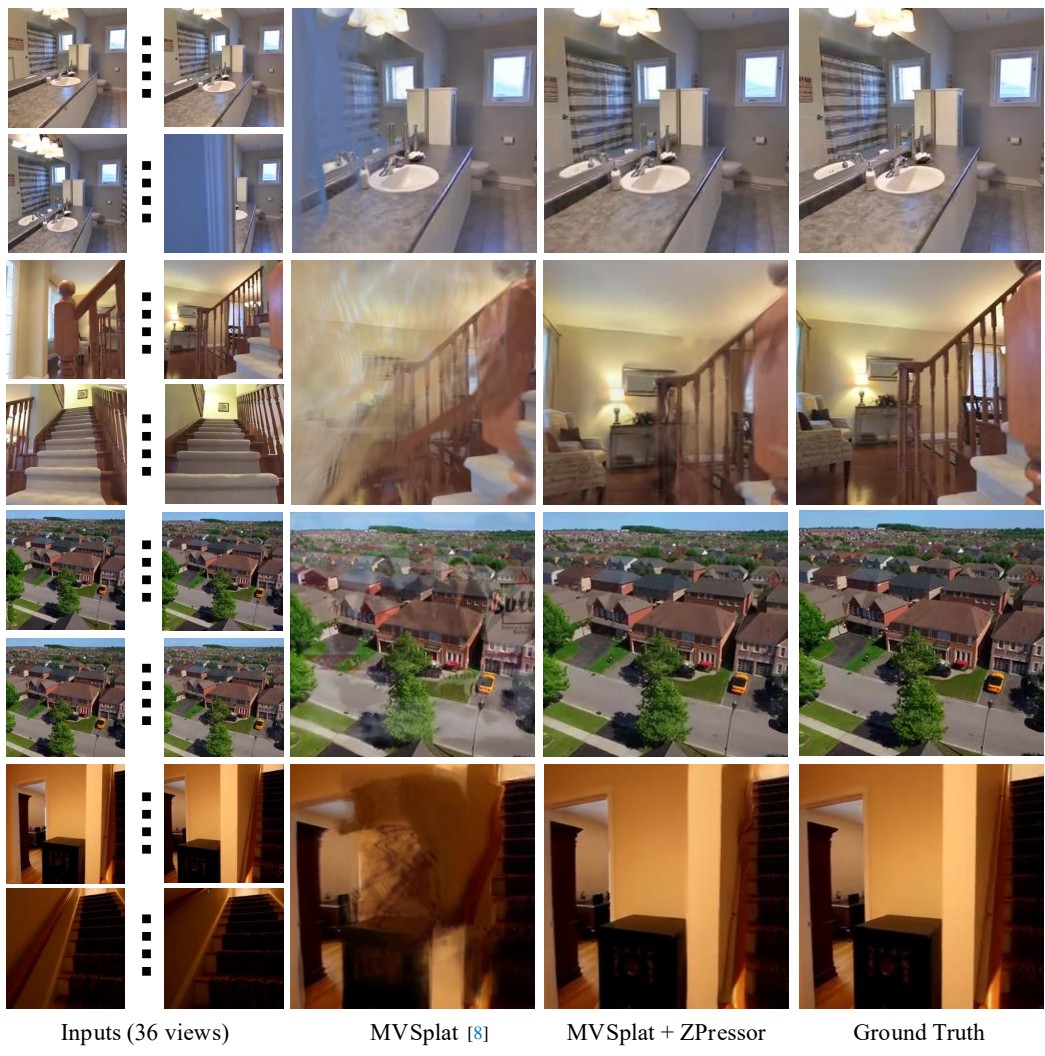

| Inputs (36 views) | MVSplat [8] | MVSplat + ZPressor | Ground Truth |

Figure F: **More qualitative comparisons on RE10K [21] with MVSplat [8] under 36 input views.** Models with ZPressor performs the best in all cases.

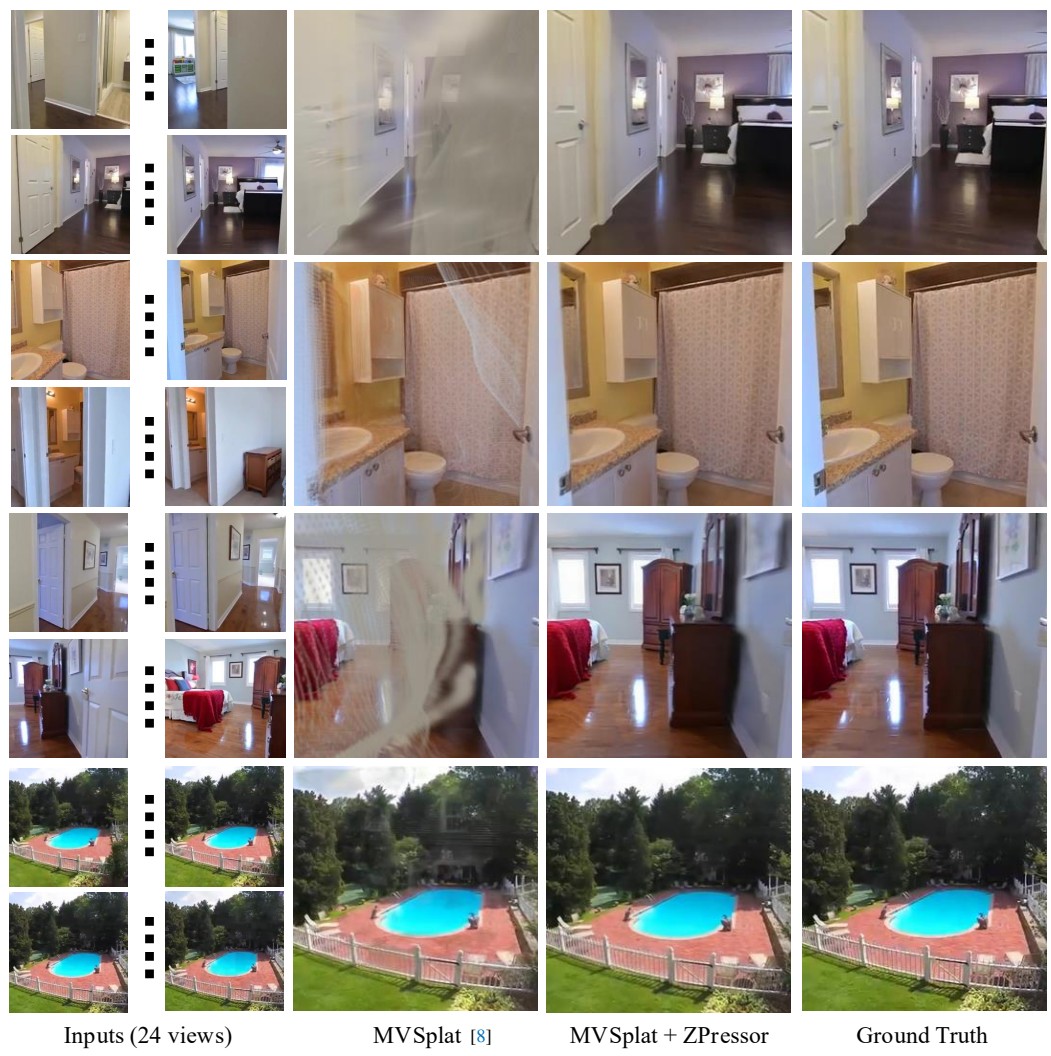

| Inputs (24 views) | MVSplat [8] | MVSplat + ZPressor | Ground Truth |

Figure G: **More qualitative comparisons on RE10K [21] with MVSplat [8] under 24 input views.**

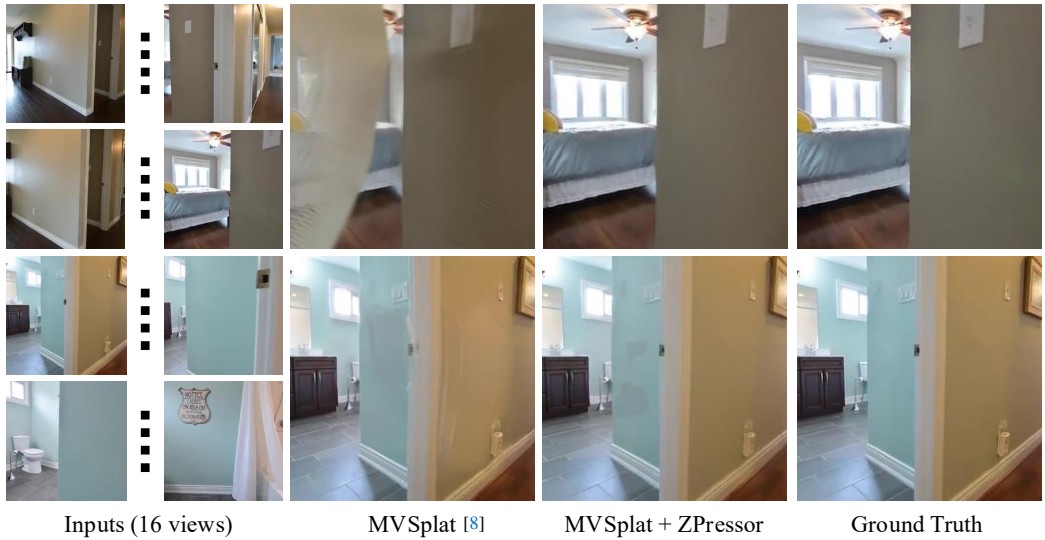

| Inputs (16 views) | MVSplat [8] | MVSplat + ZPressor | Ground Truth |

Figure H: **More qualitative comparisons on RE10K [21] with MVSplat [8] under 16 input views.**

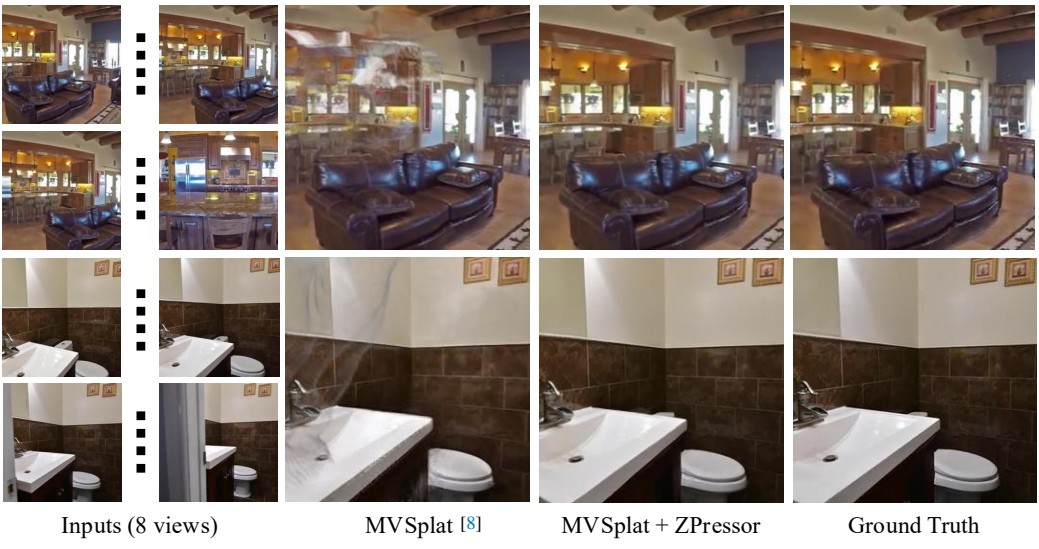

| Inputs (8 views) | MVSplat [8] | MVSplat + ZPressor | Ground Truth |

Figure I: **More qualitative comparisons on RE10K [21] with MVSplat [8] under 8 input views.**

