# OpenReview forum: "ZPressor: Bottleneck-Aware Compression for Scalable Feed-Forward 3DGS"
_NeurIPS.cc/2025/Conference — NeurIPS 2025 poster_

### Official Review · Reviewer_qeXk · 2025-06-29

**Clarity:** 2
**Significance:** 3
**Originality:** 3
**Rating:** 4
**Confidence:** 5

**Summary:**

The paper revisits feed-forward 3DGS prediction from an information bottleneck perspective and introduces ZPressor, a plug-and-play module that can improve the model performance and reduce inference/training memory. The method includes group-based compression with selected anchor views. The authors verified the proposed ZPressor on multiple base models and datasets.

**Questions:**

In general, I believe this is an interesting paper, which improving the feed-forward 3DGS reconstruction by revisiting the information bottleneck theory. However, based on the current performance patterns, biased baseline choice, and the unclear claims, I will give a boarderline reject. I encourage the authors to include more analysis regarding my questions listed in the weakness section. And I would be happy to increase my score if it is reasonable to me.

**Ethical Concerns:**

["NO or VERY MINOR ethics concerns only"]

**Final Justification:**

Thanks for the authors for the rebuttal! Previously I misunderstood the problem setting and thought it leads to worse performance by training on more images, and the authors clarified that it was using more input images during test time.
Then I think this makes a lot more sense and I will increase my score. At the same time, I would still recommend training models with global attentions to further verify the claim.

**Limitations:**

Please see the sections above.

**Quality:**

3

**Strengths And Weaknesses:**

Strengths:
- The paper revisits feed-forward 3DGS reconstruction from an information bottleneck perspective, which recalls me of the discussion why do we need inductive bias for 3D. And ZPressor is a kind of inductive bias of compression. This makes a lot of sense to me, as the essence of 3D reconstruction is just compression (of the repetitive and redundant information).
- Based the results shown in the paper, the proposed ZPressor is effective and efficient, which improves performance and reduces computation at the same time.
- The authors verified the effectiveness of ZPressor on multiple base model and datasets.


Weaknesses:
- However, at the same time, I do have some concerns about the claims and performance patterns of the baseline models of the paper. And I would like to get more feedback from the authors. Please see details below.
- First, the performance in Table 1 and Table 2 is weird to me. The results show that **with more input views, the performance drop**. The authors have also mentioned it multiple times in the writing, e.g., Line 4 and Line 31. However, this is counter-intuitive. Personally, I have never observed any performance pattern like this in my own experiments. And this is not the performance pattern of every paper that I read. For example, in Long-LRM [1], MegaSynth [2], the performance improved with more input views. I encourage the authors to explain a bit more on this.
- Second, my conjecture is that, the weird performance pattern is because of a biased base model choice. Concretely, the base models, pixelSplat, DepthSplat and MVSplat, are based on pairwise computation, e.g., cost volume calculation; while the two paper I mentioned above adopts global attentions on all views, which has less inductive bias in model architecture design. It would be beneficial to explore whether this assumption is true by adapting ZPressor on models with global attention.
- Third, in Table 2 and 3, after adding the ZPressor module, the performance still drop with more input images. Thus, this experimental result cannot support the claim of the author on solving the ``degraded performance'' problem.
- Forth, the authors mentioned *limited capacity of the encoder* multiple times. However, I am not able to understand this claim. What exact ``capacity'' the authors were referring to here?
- Finally, the visualization examples in the supplementary videos is in very low quality to me. In general, it is weird that the performance is so bad and the renderings are so blurry.

[1] Chen, Ziwen et al. “Long-LRM: Long-sequence Large Reconstruction Model for Wide-coverage Gaussian Splats.” ArXiv 2024.
[2] Jiang, Hanwen et al. “MegaSynth: Scaling Up 3D Scene Reconstruction with Synthesized Data.” CVPR 2025.

---

> ### Author Rebuttal · Authors · 2025-07-31
>
> ## **Response to Reviewer qeXk**
>
> We sincerely thank Reviewer qeXk for the thoughtful review and valuable feedback. We have carefully considered all points raised and provide our responses in detail below:
>
> ---
>
> ### **W1. Performance drop with more input views in Tables 1 and 2**
>
> 1. The numerical values presented in the paper for models such as Long-LRM and MegaSynth mentioned by the reviewer are **based on fine-tuning (e.g., Long-LRM Section 4.2)/re-training (e.g., MegaSynth Section 6.5) for each test input**. However, during training, we used 12 input views as input, and during testing, we directly applied inputs with more than 12 input views to our models **for generalization**. While the model performance may slightly decrease, it is still significantly better than the substantial decline observed in the baseline models.
> 2. Furthermore, these baseline models exhibit redundancy under multi-view input conditions, which inherently leads to performance degradation when the number of views exceeds the model's inherent capabilities. Consistent with the results in Table 1 of PixelGaussian [1], we also observed test degradation in feed-forward models when increasing the number of input views during the testing phase.
>
> ---
>
> ### **W2. Need for models with global attention**
>
> We thank the reviewer for this insightful comment. We agree that our selected based models (pixelSplat, DepthSplat, MVSplat) rely on pairwise computations, which carry a different inductive bias compared to the global attention mechanisms in recent works like Long-LRM and MegaSynth. We are keen to test ZPressor on global attention models. However, we were unable to do so as Long-LRM lacks an official implementation and we failed to reproduce results from the community version. Similarly, MegaSynth has not yet released its model implementation.
>
> We agree that global attention is a promising direction for dense-view reconstruction, which motivates the design of Long-LRM and MegaSynth. Nevertheless, our primary contribution is to enhance existing sparse-view reconstruction methods for dense-view scenarios (L63), and it was not our intention to imply that our module would yield similar improvements for models already architected for dense-view reconstruction. We will clarify this in the updated version and explore global attention-based architectures in future work.
>
> ---
>
> ### **W3. Performance still drops with ZPressor for more views**
>
> As addressed in W1, we didn't finetune for every input view, so a reasonable performance drop is observed.
>
> We would like to clarify that we do not claim to "solve" the "degraded performance" problem. Instead, our work focuses on **mitigating** this issue. Our approach does not offer a perfect solution for generalization (beyond fine-tuning) to extremely dense views in feed-forward scene reconstruction. We are indeed the pioneering work in dealing with this important problem. And we will clarify this point in the updated version of our paper.
>
> ---
>
> ### **W4. "Limited capacity of the encoder" unclear**
>
> The **capacity** of an image-to-3DGS network refers to its ability to effectively process redundant information across dense views without leading to degraded performance or memory issues.
>
> The phrasing "limited encoder capacity" in our discussion may be misleading. Our intention was to refer to the network from image input to 3D Gaussian prediction.To prevent misinterpretation, we will address this ambiguity in an updated version of our paper by rephrasing "encoder" as "the whole network from image to 3DGS".
>
> ---
>
> ### **W5. Low quality of visualization examples**
>
> 1. Due to GPU resource constraints, we adopted the configurations of DepthSplat, MVSplat, and pixelSplat. This resulted in 480P visualization, which can make the rendering appear blurry. Figure 3 and the supplementary videos in our paper used the same rendering method, so perhaps scaling the video playback window to an appropriate size could achieve a suitable playback effect.
> 2. The baseline models perform poorly with 36-view inputs, but that adding ZPressor significantly improves their performance.
> 3. We are finetuning the ZPressor-integrated models on 960P datasets and will include the results in a updated version.
>
> ---
>
> ### Reference
>
> [1] Xin Fei et al. “PixelGaussian: Generalizable 3D Gaussian Reconstruction from Arbitrary Views.” Arxiv 2024.

---

> ### Author Response · Authors · 2025-08-04
>
> Dear Reviewer qeXk,
>
> Did we satisfactorily answer your questions? Would you like us to clarify anything further? Feel free to let us know, many thanks.
>
> Best regards,
>
> Authors of #3951

---

> > ### Comment · Reviewer_qeXk · 2025-08-04
> >
> > Thanks for the rebuttal! I have increased my score.

---

> > > ### Author Response · Authors · 2025-08-05
> > >
> > > Dear Reviewer qeXk,
> > >
> > > Thank you very much for your thoughtful feedback and for reconsidering your evaluation of our work. We are grateful for your positive comments and for raising the score.
> > >
> > > Your insights are highly valuable, and we will be sure to revise and improve the manuscript accordingly in the updated version.
> > >
> > > Thank you again for your time and constructive engagement.
> > >
> > > Best regards,
> > >
> > > Authors of #3951

---

### Official Review · Reviewer_8gSJ · 2025-06-30

**Clarity:** 4
**Significance:** 3
**Originality:** 4
**Rating:** 5
**Confidence:** 5

**Summary:**

This paper introduces ZPressor, a lightweight, architecture-agnostic module designed to enhance the scalability of feed-forward 3D Gaussian Splatting (3DGS) models, which are prominent for real-time novel view synthesis. Traditional 3DGS approaches struggle to efficiently process a large number of input views due to their limited encoder capacity, resulting in degraded performance and high memory usage as the number of views increases. The authors analyze these limitations through the lens of the Information Bottleneck (IB) principle, which emphasizes the preservation of task-relevant information while discarding redundancy. Inspired by this, ZPressor employs a strategy where input views are divided into anchor views and support views. Cross-attention mechanisms fuse the support view information into anchor views, compressing the dense multi-view data into a compact latent representation Z that retains essential scene details. By integrating ZPressor into existing feed-forward 3DGS models, they demonstrate significant improvements: models can handle over 100 dense input views at high resolution (480P) on standard GPUs without performance drops or excessive memory consumption. Their experimental validation on large-scale benchmarks (like DL3DV-10K and RealEstate-10K) shows consistent enhancements in quality and efficiency, making 3D scene reconstruction more practical for real-world applications.

**Questions:**

- In the experiments, how do you split input views and target novel views?
- Do you select the same target novel views for different numbers of input views?
- For anchor view selection, did you try other strategies? e.g., k-means clustering.

**Ethical Concerns:**

["NO or VERY MINOR ethics concerns only"]

**Final Justification:**

After reading the comments from other reviewers and the rebuttal from the authors, I would like to keep my score for acceptance.

**Limitations:**

Limitations are mentioned in weaknesses.

**Paper Formatting Concerns:**

There are no paper formatting concerns.

**Quality:**

4

**Strengths And Weaknesses:**

**Strengths:**
- This paper is well-written, motivations and significance of the task are clearly described in the paper.
- The use of the Information Bottleneck principle provides a solid theoretical foundation for addressing the limitations of existing 3DGS models, offering insights into why they struggle with dense views and how to mitigate these issues.
- ZPressor is designed to be architecture-agnostic, allowing it to be seamlessly integrated into various existing feed-forward 3DGS models, which broadens its applicability.
- Extensive experiments on large-scale benchmarks like DL3DV-10K and RealEstate-10K demonstrate consistent performance improvements, showing that the approach is effective both qualitatively and quantitatively.

**Weaknesses:**
- The authors acknowledge that ZPressor may be less effective when dealing with extremely dense views (e.g., thousands of views), where even the compressed representations may become computationally challenging to handle.
- Incorporating cross-attention-based compression modules introduces extra complexity into the pipeline, which may affect training stability, implementation, or inference speed, although it is designed to be lightweight.

---

> ### Author Rebuttal · Authors · 2025-07-31
>
> ## **Response to Reviewer 8gSJ**
>
> We sincerely thank Reviewer 8gSJ for the positive evaluation and for recognizing the strengths of our paper, including its clear motivation, theoretical foundation, versatility, and the extensive experimental validation. We are particularly pleased that reviewer found the use of the Information Bottleneck principle insightful.
>
> ---
>
> ### **W1. Less effective when dealing with extremely dense views**
>
> We acknowledge that ZPressor's effectiveness is limited in extremely dense views (e.g., thousands of views), as detailed in L322-L327. Our future work may combine ZPressor with further 3D Gaussian merging or memory-efficient rendering techniques to tackle such scenarios.
>
> ---
>
> ### **W2. Complexity from Cross-Attention**
>
> **Inference efficiency and parameters comparison:** Below is a comparative analysis of DepthSplat before and after the integration of ZPressor, focusing on trainable parameters, inference speed, and memory consumption when processing 36 input views:
>
> | **Methods**         | **Parameters** | **Inference Time (ms)** | **Memory (GB)** |
> | ------------------- | -------------- | ----------------------- | --------------- |
> | DepthSplat          | **114M**       | 692.3                   | 17.84           |
> | DepthSplat+ZPressor | 116M           | **204.7**               | **3.59**        |
>
> As demonstrated by our results, ZPressor introduces a mere 1.7% increase in parameters while achieving significant reductions of 70% in inference time and 80% in memory consumption. This clearly underscores ZPressor's lightweight nature and its substantial contribution to enhancing inference efficiency.
>
> **Implementation:** ZPressor is designed with an architecture-agnostic approach, allowing for seamless integration and expansion of the scalability of existing feed-forward models.
>
> **Training stability:** Our experiments confirm that ZPressor does not compromise model stability during training. In fact, by assisting the model in eliminating feature redundancy, ZPressor contributes to a more stable training process.
>
> ---
>
> ### **Q1. How to split input views and target novel views**
>
> Following the standard protocol in existing feed-forward 3DGS works (e.g., DepthSplat/MVSplat/pixelSplat), for each scene in the datasets (DL3DV-10K and RealEstate10K), we select the context views by picking frames with a fixed "context gap" (e.g., skipping a certain number of frames) around the target view or a central reference frame, ensuring a diverse set of inputs.
>
> The target novel views are chosen at different, distinct camera positions (frames) that were not part of the input context views. These target views are usually selected to be spatially separated from the context views to rigorously test the model's generalization capabilities for novel view synthesis.
>
> For evaluation, we render eight fixed target novel views, as stated in Section 4.2 of the paper.
>
> ---
>
> ### **Q2. Whether to select the same target novel views for different numbers of input views**
>
> **Yes**, for a fair comparison across different numbers of input views (e.g., 8, 12, 16, 24, 36 views), we use the same set of eight target novel views for evaluation. This ensures that the performance gains or drops observed are solely attributable to the changes in the input view configuration and the effectiveness of ZPressor, rather than variations in the evaluation criteria.
>
> ---
>
> ### **Q3. Other anchor frame selection strategy**
>
> We experimented with **K-Means clustering** on camera positions, and the results are presented in the table below.
>
> Furthermore, we explored **feature-based anchor view selection strategies**. By clustering view features using K-Means to obtain view groups, and then calculating the anchor view, ZPressor can support pose-independent anchor selection. The experimental results for this approach are also included in the table below.
>
> | **Views** | **Methods**                           | **PSNR**  | **SSIM**  | **LPIPS** |
> | --------- | ------------------------------------- | --------- | --------- | --------- |
> | 36        | DepthSplat                            | 19.23     | 0.666     | 0.286     |
> | 36        | DepthSplat+ZPressor(w/o pose+K-Means) | 22.81     | 0.791     | 0.174     |
> | 36        | DepthSplat+ZPressor(w/ pose+K-Means)  | 22.84     | 0.789     | 0.175     |
> | 36        | DepthSplat+ZPressor                   | **23.88** | **0.815** | **0.150** |

---

> > ### Comment · Reviewer_8gSJ · 2025-08-05
> > **Thanks for the rebuttal**
> >
> > All my questions and concerns are addressed by the authors. I would like to keep my score and champion the paper for acceptance.

---

> > > ### Author Response · Authors · 2025-08-05
> > >
> > > Dear Reviewer 8gSJ,
> > >
> > > Thank you for your feedback and for confirming that we have addressed your concerns.
> > >
> > > We sincerely appreciate your consistently high evaluation and, in particular, your generous offer to champion the paper. Your support is immensely encouraging.
> > >
> > > Thank you for your invaluable contribution.
> > >
> > > Best regards,
> > >
> > > Authors of #3951

---

> ### Author Response · Authors · 2025-08-04
>
> Dear Reviewer 8gSJ,
>
> Did we satisfactorily answer your questions? Would you like us to clarify anything further? Feel free to let us know, many thanks.
>
> Best regards,
>
> Authors of #3951

---

### Official Review · Reviewer_cRJd · 2025-07-01

**Clarity:** 2
**Significance:** 2
**Originality:** 3
**Rating:** 4
**Confidence:** 4

**Summary:**

The author tackles the view scalability problem of the modern feedforward based 3DGS method. They find that with increasing number of views,  methods like MVSplat, pixelSplat, DepthSplat struggle with reconstruction quality. The author attributes such degradation to the "limited capacity of the encoder". Hence, motivated by information bottleneck theory, they propose a plug-and-play model ZPressor that dramatically reduce the input features from image encoder and generate enhanced visual quality.

**Questions:**

- confusion definition of "variational rendering loss" at the end of section  3.
- what do you mean by "information overload" in section 4.2? intuitively, those information from different views should be redundant, what does overload mean here and why it leads to performance degradation then?
- since the current baseline is too weak, plz **demonstrate your module on a feedforward 3DGS methods that natively support input views more than 6**. Recent works, for example anysplat, FLARE, InstantSplat, GPSGaussian++, and any other method are acceptable. Or you can simply choose to fine-tune MVSplat/PixelSplat/DepthSplat with multi views as the baseline. Don't have to stick with single 80GB GPU constraints.

**Ethical Concerns:**

["NO or VERY MINOR ethics concerns only"]

**Final Justification:**

Author has provided further experiment results to demonstrate their claim. My concerns are resolved.

**Limitations:**

It relies on known cam pose for selecting anchor views. Yet today's sota work could also works on poes-free inputs (noposplat, anysplat, PF3plat). How your method could help in this scenario ?

**Paper Formatting Concerns:**

Null

**Quality:**

3

**Strengths And Weaknesses:**

Good part:
- the paper is well written and easy to follow with abundant visual results.
- the identified problem is a true pain for sparse view feedforward 3DGS prediction and leveraging cross attention for information fusion is generally a plausible solution
- the module as a feature digester does reduce the hardware overhead of previous methods and achieves remarkable enhancement.

Weaknesses:
- the key claim of the paper is that "limited capacity of encoder" as the main drawback of previous methods. Yet for all three methods they have modified are SINGLE VIEW image encoder which means that no information has been exchanged between views during the encoder stage. **That's to say, the key reason of bad performance of previous methods is not the encoder**
- the key reason of poor performance of those methods are very simple actually: they are all trained on very sparse views typically 2-4, with depthSplat at most 6 views. Yet the evaluated views of ZPressor is 8~36, which falls far outside the distribution of training data of those methods. **That's to say: the baselines are simply too weak, and are not expected to handle the dense view scenario**.
- The author use cross attention to fuse information from support views to anchor views, yet the ablation study of tab 3 shows very limited enhancement from anchor-fusion to support fusion. Considering the proposed method is trained with 6+6 views, **such enhancement could simply be attributed to the model's getting familiar with more views, instead of learning extra information from support views.**
- It is not supposed to claim a method as `plug-and-play` if you integrate your module and still need to train the integrated new model from scratch.
- IB theory is a little bit out-of-place here.

---

> ### Author Rebuttal · Authors · 2025-07-31
>
> ## **Response to Reviewer cRJd**
>
> We are thankful to Reviewer cRJd for their insightful comments. Our detailed responses to the points are as follows:
>
> ---
>
> ### **W1. "Limited capacity of encoder" definition**
>
> We agree that our phrasing of "limited capacity of the encoder" might be misleading. Our intention was to refer to the entire network from image input to 3D Gaussian prediction (as mentioned in L27), which acts as a **“scene encoder”** by extracting scene-dependent features and predicting 3D Gaussians in a single forward pass.
>
> We will clarify this point in the updated version of the paper by rephrasing "encoder" to "the whole network from image to 3DGS" to avoid confusion.
>
> ---
>
> ### **W2. Baselines are "too weak" for dense views**
>
> Thanks for reviewer's insightful comments regarding the performance of our baselines and the suggestion to evaluate ZPressor on feed-forward 3DGS methods that natively support more input views. We appreciate the opportunity to clarify these points.
>
> 1. We did not use the original settings or pre-trained weights of MVSplat, PixelSplat, and DepthSplat. Instead, we **re-trained** these baseline models from scratch under multi-view settings, adhering to the training strategies described in their respective papers while adjusting for multi-view inputs where applicable. This consistent training approach for both the baselines and our ZPressor-integrated models is reflected in Tables 1 and 2. This ensures that the observed improvements are due to ZPressor's bottleneck-aware compression and not merely a result of providing more input views to models in training stage.
> 2. The existing baselines (pixelSplat/MVSplat/DepthSplat) are the **best options** we have in this submission. Reviewer suggested evaluating on more recent feed-forward 3DGS methods like AnySplat, FLARE, InstantSplat, and GPSGaussian++. However, there are some practical limitations: (1) AnySplat was released on a arXiv (29 May 2025) *after* NeurIPS submission deadline. (2) FLARE’s official implementation does not provide readily available NVS training code. (3) InstantSplat requires approximately 1000 iterative optimizations, which is fundamentally different from the single-forward-pass nature of the feed-forward 3DGS models we target. (4) GPSGaussian+ primarily focuses on free-viewpoint video synthesis of human characters.
>
> Covering every conceivable baseline would extend far beyond the scope of our paper and incur prohibitive resource costs. Our module's core functionality—compressing multi-view features—is not confined to any specific model, and its demonstrated benefits across the chosen baselines adequately showcase its broad applicability and contribution.
>
> ---
>
> ### **W3. Limited enhancement from "anchor-fusion to support-fusion" in Tab. 3**
>
> The key to understanding this lies in the nature of the information contributed by the support views. The support views don't introduce entirely new macro-level information, but rather **enhancing the richness and consistency of details**, its impact on broad numerical metrics like PSNR might be less dramatic.
>
> 1. When we consider a baseline of 6 anchor views, these views, selected through Farthest Point Sampling, already provide substantial spatial coverage of the scene. Therefore, they capture the primary structural and appearance information. The "support views," while adding to the overall number of input views, are by definition spatially closer to existing anchor views.
> 2. Their primary contribution is not to drastically improve evaluation results but rather to provide finer-grained, complementary details or to reinforce information in areas already partially observed by the anchor views. This additional information is often concentrated on subtle aspects of the scene.
>
> ---
>
> ### **W4. Not "plug-and-play" if training from scratch**
>
> We'll clarify this nuance in the updated version and will no longer refer to our module as "plug-and-play" as this was an unintentional misleading.
>
> Our initial intention was for our model to be **architecture-agnostic**. While fine-tuning or training from scratch on the combined architecture is typically required to fully leverage ZPressor's benefits, the architectural integration itself is straightforward.
>
> ---
>
> ### **W5. IB theory "out-of-place"**
>
> We believe the IB principle provides a robust theoretical foundation and a powerful guiding framework for our work. Inspired by the Information Bottleneck principle, we analyzed the information flow within the Feed-Forward 3DGS model. This led us to propose learning a compact and informative feature representation that discards redundancy while retaining sufficient original information. This aligns perfectly with the IB objective: finding the minimal sufficient statistic of the prediction network's input to predict its output.
>
> ---
>
> ### **Q1: Confusion definition of "variational rendering loss"**
>
> The sentence in Line 202 "we optimize the feed-forward 3DGS with ZPressor by the variational rendering loss, e.g., MSE and LPIPS" may be misleading. Our variational loss consists of two main components: a reconstruction term (e.g., MSE and LPIPS) and a compression term (calculated as a KL divergence weighted by β). So, what we mean to say is that the variational loss **is based on** the rendering loss.
>
> ---
>
> ### **Q2. "Information overload" in Section 4.2**
>
> For many feed-forward 3DGS models, as input views become dense, the network is overwhelmed by the massive, redundant information from highly overlapping regions, leading to an information overload. This information overload arises because the network **lacks the capacity to effectively process the massive amount of redundant information** across dense views, leading to degraded performance and memory issues. This can particularly impact generalization during inference with more views.
>
> ZPressor tackles this challenge by acting as a "bottleneck-aware" module within the scene encoding process. It explicitly manages this multi-view information flow by compressing view features, thereby reducing redundancy and distilling essential information to overcome the "information overload.”
>
> ---
>
> ### **Q3. Demonstrate module on methods supporting more input views**
>
> Please refer to our response in W2 for more clarification.
>
> ---
>
> ### **Limitation. Reliance on known camera poses**
>
> We appreciate the reviewer for highlighting this important aspect. For the works that can handle pose-free inputs, NoPoSplat is based on the MASt3R architecture and pre-trained weights, was only trained on two views, making it an unfair comparison to use as a baseline; AnySplat was released on a arXiv (29 May 2025) after NeurIPS submission deadline; PF3plat explicitly predict the poses, which is essentially no different from models with pose input. For these reasons, we did not apply ZPressor to these models at the beginning.
>
> We also explored a pose-free ZPressor compression method. This approach leverages K-Means clustering on the features of input views to identify distinct view groups. Subsequently, an anchor view is determined for each group, enabling a pose-independent anchor view selection process.
>
> We conducted preliminary validation on pose-free setting using DepthSplat as our baseline:
>
> | **Views** | **Methods**                   | **PSNR**  | **SSIM**  | **LPIPS** |
> | --------- | ----------------------------- | --------- | --------- | --------- |
> | 36        | DepthSplat                    | 19.23     | 0.666     | 0.286     |
> | 36        | DepthSplat+ZPressor(w/o pose) | 22.81     | 0.791     | 0.174     |
> | 36        | DepthSplat+ZPressor(w/ pose)  | **23.88** | **0.815** | **0.150** |
>
> The experimental findings clearly indicate ZPressor's substantial performance lead over the baseline, even when foregoing camera poses, which boosts the scalability of feed-forward models.

---

> ### Author Response · Authors · 2025-08-04
>
> Dear Reviewer cRJd,
>
> Did we satisfactorily answer your questions? Would you like us to clarify anything further? Feel free to let us know, many thanks.
>
> Best regards,
>
> Authors of #3951

---

> ### Comment · Reviewer_cRJd · 2025-08-05
>
> The key reason i post a weak rejection for this paper is that: Ive got a strong feeling of distancing between this paper's theory and method. You have got several specious claim and concepts in both your original paper and your rebuttal, including:
> - **The only part your method has a relationship with your IB theory is the KL divergence reg term** which you just told me, while there is no formal definition in the paper (i know what it is, i just cannot find it anywhere in the paper). Moreover, **there is no ablation study to demonstrate its effectiveness and contribution**.
> - you have defined "Information overload" as "the network lacks the capacity to effectively process the massive amount of redundant information". Actually, people with info theory background will never call information redundancy as "information overload". That is simply representation overload or message overload. Or when you make such claim, do you implicit means that the network actually can handle dense but distinct views as long as they are not redundant. Or do you any evidence that you believe the failure for feedforward reconstruction is caused by those "redundant information" only?
> - Your rebuttal over W3 still has the same problem: you claimed that "Their primary contribution is not to drastically improve evaluation results but rather to provide finer-grained, complementary details or to reinforce information in areas already partially observed by the anchor views. " but sill no experimental demonstration, at least another numeric metrics that could demonstrate what you just claim to me.
>
> if you do insist the significance of your IB theory, i would recommend you to have a reference of 3DGS-MCMC who shows an excellent example of theory guided method design.
>
> Personally i am no a fan of solving a simple problem with complicated theory. The issue you point out is a true pain for ff 3dgs, thats the good part. Yet if we could solve it with straightforward methods like pre-filtering the views so that it could be fed to previous sparse view models, or simply using those multi-view compatible architectures, why not do it?

---

> > ### Author Response · Authors · 2025-08-06
> > **Further response to follow-up comments (1/2)**
> >
> > We sincerely thank Reviewer cRJd for the continued engagement and for acknowledging that the view scalability problem we address is a "true pain for ff 3dgs." We would like to address your key concerns below.
> >
> > ## **Relationship with IB Theory**
> >
> > 1. **ZPressor is a straightforward method in a technical perspective:** ZPressor is essentially a learnable approach to *pre-filtering* and *fusion*, aiming to connect dense inputs with existing sparse-view models. A naive pre-filtering that simply discards views risks losing critical information, as some views might contain additional unique contents of the scene. In contrast, our method *pre-filter* certain views as anchors and *fuses* the remaining views into them. This process yields a compact representation that retains *maximum scene coverage* and detail.
> > 2. **ZPressor is inspired by the IB theory in a concept perspective:** We take inspiration from two key aspects of the IB theory: the concept of minimal sufficient latent representation and the loss function structure. (I) The minimal sufficient latent representation concept motivates us to introduce a bottleneck at the feature level, leading to an implementation of Farthest Point Sampling (ensure minimal) with Cross-Attention (ensure sufficient). (II) The loss functions, with its compression and reconstruction terms, provides a clear blueprint for supervising this process. The formulation can be denoted as:
> >
> > $$
> > \begin{equation} \mathcal{L} = \underset{\mathcal{Z} \sim p_\theta(\mathcal{Z} \mid \mathcal{X})}{\mathbb{E}} \bigl[ -\log q_\phi(\mathcal{Y} \mid \mathcal{Z}) \bigr]  + \beta \underset{\mathcal{X}}{\mathbb{E}} \bigl[ \text{KL} \bigl[ p_\theta(\mathcal{Z} \mid \mathcal{X}), r(\mathcal{Z}) \bigr] \bigr] \end{equation}
> > $$
> >
> > where $\phi$ denotes the parameters of 3D Gaussians prediction network and $-\log q_\phi(\mathcal{Y} \mid \mathcal{Z})$ can be modeled by MSE/LPIPS loss, $\theta$ denotes the parameters of the network preceding $\mathcal{Z}$ and $p_\theta(\mathcal{Z} \mid \mathcal{X})$ is the posterior probability estimate of $\mathcal{Z}$, $r(\mathcal{Z}) \sim \mathcal{N}(\mathcal{Z} \mid \mu_0, \Sigma_0)$ is the Gaussian prior of $\mathcal{Z}$.  During training, we use the reparameterization trick (Kingma & Welling, 2014) [1] to estimate the gradients, following VIB [2] and VIBERT [3].
> >
> > Our ablation study results for the compression term are as follows:
> >
> > | **Views** | **Methods**                                             | **PSNR**  | **SSIM**  | **LPIPS** |
> > | --------- | ------------------------------------------------------- | --------- | --------- | --------- |
> > | 36        | DepthSplat                                              | 19.23     | 0.666     | 0.286     |
> > | 36        | DepthSplat + ZPressor (w/o compression term, $\beta=0$) | 23.43     | 0.806     | 0.165     |
> > | 36        | DepthSplat + ZPressor ($\beta=10^{-5}$)                 | **23.88** | **0.815** | **0.150** |
> >
> > Experiments have proven that the KL term can improve the quality of compressed features, resulting in better rendering results.
> >
> > In the revision, we will incorporate the relevant discussions and ablation study.
> >
> > ## **Clarification of "information overload"**
> >
> > We agree that "representation overload" is a more accurate term. Our intention was that the capacity of "the whole network from image to 3DGS" is overwhelmed by the feature tokens from dense views. This leads to a degradation in the network's ability to effectively distill a coherent 3D scene representation.
> >
> > We will replace all instances of "information overload" with "representation overload" and add a brief explanation to clarify that we are referring to the architectural bottleneck of processing an excessive number of feature tokens in the updated version.
> >
> > ## **Numerical Improvement of "support fusion" (W3)**
> >
> > While the benefits of support view fusion are most striking visually (which we unfortunately cannot demonstrate in this text format), we have reported a new experimental results to quantitatively highlight its importance.
> >
> > The rationale is as follows: the contribution of a support view becomes most critical when the target (novel) view is spatially close to support view, rather than an anchor view. In this scenario, fusing the support view should yield a significant advantage:
> >
> > | **Methods**                            | **PSNR**  | **SSIM**  | **LPIPS** |
> > | -------------------------------------- | --------- | --------- | --------- |
> > | DepthSplat + ZPressor (anchor fusion)  | 23.61     | 0.799     | 0.137     |
> > | DepthSplat + ZPressor (support fusion) | **24.97** | **0.836** | **0.161** |
> >
> > These results show a much larger performance gap (+5.77% PSNR/+4.55% SSIM/-14.86% LPIPS) when evaluating views where the support views is most relevant, confirming that our fusion mechanism effectively works.
> >
> > We will add visualization and ablation results to the updated version to explain this issue more clearly.
> >
> > *(see next comments for more response, thanks)*

---

> > ### Author Response · Authors · 2025-08-06
> > **Further response to follow-up comments (2/2)**
> >
> > ## **Conclusion**
> >
> > Our main contribution is a technical solution to the dense-view challenge in feed-forward 3DGS models. We introduce an architecture-agnostic module, ZPressor, that enables sparse-view models to handle dense-view inputs. To our knowledge, ZPressor is the first approach to directly adapt sparse-view models for dense-view scenarios in this way.
> >
> > In implementing ZPressor, we **drew inspiration from IB** theory, compressing the scene into a minimal sufficient latent representation and training the model with an IB loss. However, **we agree that ZPressor should be viewed primarily as a practical technical module, and while it is empirically consistent with the IB theory,  it is not our intention to claim that it is a mathematically rigorous application of IB theory.** In our updated version, we will clarify this positioning, emphasize our technical contribution, and tone down the significance of IB theory.
> >
> > Thank you once again for recognizing the significance of the issue we are addressing. We will release our code and pretrained weights for all relevant models. We hope that ZPressor will serve as a valuable and practical resource for the community.
> >
> > Please let us know if there is anything else we can clarify.
> >
> > ## Reference
> >
> > [1] Diederik P Kingma et al. “Auto-Encoding Variational Bayes.” ICLR 2014.
> >
> > [2] Rabeeh Karimi Mahabadi et al. “Deep Variational Information Bottleneck.” ICLR 2017.
> >
> > [3] Rabeeh Karimi Mahabadi et al. “Variational Information Bottleneck for Effective Low-Resource Fine-Tuning.” ICLR 2021.

---

> > > ### Comment · Reviewer_cRJd · 2025-08-06
> > >
> > > Concerns are resolved and my intention isn’t to be harsh. Will raise my score.

---

> > > > ### Author Response · Authors · 2025-08-06
> > > >
> > > > Dear Reviewer cRJd,
> > > >
> > > > Thank you for the positive feedback. We are grateful for your support and for recognizing the value of our work.
> > > >
> > > > We will integrate our discussions into the updated version to further strengthen our contribution and avoid unintentional misleading.
> > > >
> > > > Thank you again for your time and insightful suggestions.
> > > >
> > > > Best regards,
> > > >
> > > > Authors of #3951

---

### Official Review · Reviewer_nH8F · 2025-07-01

**Clarity:** 3
**Significance:** 2
**Originality:** 2
**Rating:** 4
**Confidence:** 4

**Summary:**

Gaussian Splatting reconstruction has recently become a popular approach for feed-forawrd reconstruction. However, many approaches reconstruct the scene by predicting Gaussian Splats from every image, which becomes memory intensive when the number of input frames is high. This paper addresses this issue with an idea inspired by the information bottleneck principle: effectively compressing a long sequence of frames into a latent representation that operates on a smaller set of keyframes.

**Questions:**

Eq 6: I think the function operating on $Q, K, V$ is attention and not Cross-attn. What makes this cross attention is the source of $K$ and $V$.

**Ethical Concerns:**

["NO or VERY MINOR ethics concerns only"]

**Final Justification:**

After reading the rebuttal, the other reviews, and additional responses, most of my concerns are sufficiently addressed to recommend acceptance.

**Limitations:**

Yes

**Quality:**

3

**Strengths And Weaknesses:**

## Strengths
- Compression for Gaussian Splatting representations is an important topic that is currently undexplored.
- The paper is well written and good to follow. The notation is clear and precise.
- The qualitative and quantitative results show that the method is effective at compression and thus can eliminate some floater artefacts. This, in turn, results in better reconstruction scores.

## Weaknesses
There is a disconnect between the theoretical motivation through the information bottleneck and the actual implementation. Line 202 reads “we optimize the feed-forward 3DGS with ZPressor by the variational rendering loss, e.g., MSE and LPIPS.”. In my understanding, MSE is not a variational loss. It can be part of a variational loss if used as a reconstruction loss in a VAE-like formulation, but this would require a KL term. Similarly, while the framework is motivated via Eq. (3), but since there is no explicit term measuring the compression score, the hyperparameter $\beta$ is unused, and no balance between compression and reconstruction is learned. It is, in a sense, pre-defined by the architecture choice.
The IB connection does not really help in designing, training, or improving the model .

The choice of anchor frames is purely based on the camera position. Typically, in SLAM-like applications, the view angle is taken into account when identifying new keyframes. This could be done in many ways, for example by computing the overlap in view frustrums. This clustering is chosen ad-hoc, and likely needs to be manually tuned. Can the IB formulation be used to learn the anchor frame selection strategy?

MV Splat associates a confidence with every Gaussian. A simple baseline would use the confidence to prune Gaussians for compression. Would that reduce the floater artefacts present in the original MVSplat reconstructions?

---

> ### Author Rebuttal · Authors · 2025-07-31
>
> ## **Response to Reviewer nH8F**
>
> We are grateful to Reviewer nH8F for the thorough evaluation of our work and for providing valuable suggestions. We have carefully considered all comments and provide our responses below:
>
> ---
>
> ### **W1. Clarification of the Information Bottleneck principle and implementation**
>
> We appreciate the reviewer’s comment and suggestion. Our initial intention is that variational loss is **based on** MSE or LPIPS. Specifically, the loss implementation includes a first term based on MSE or LPIPS, and a second term based on a KL divergence multiplied by a coefficient β. These correspond to the reconstruction and compression terms, respectively.
>
> This formulation follows the Information Bottleneck (IB) principle, which aims to learn a compact latent representation that retains task-relevant information while discarding irrelevant details. The reconstruction term encourages preservation of useful information for the output task, while the KL divergence penalizes excessive mutual information between the input and latent variables, thus enforcing compression. By adjusting the coefficient β, we control the trade-off between fidelity and compactness, balancing the model’s expressiveness and generalization.
>
> We will clarify this unintentional misleading in the updated version.
>
> ---
>
> ### **W2. Anchor frame selection strategy (View Overlap)**
>
> We appreciate the reviewer's valuable suggestions. We agree that incorporating view geometry (like view frustum overlap) could lead to more optimal anchor selection. Our current approach, farthest point sampling (FPS) based on camera positions, is a simple yet effective heuristic to ensure spatial diversity among anchor views. We report a comparison of experimental results using an anchor view selection method based on the overlap in view frustums:
>
> | **Views** | **Methods**                        | **PSNR**  | **SSIM**  | **LPIPS** |
> | --------- | ---------------------------------- | --------- | --------- | --------- |
> | 36        | DepthSplat                         | 19.23     | 0.666     | 0.286     |
> | 36        | DepthSplat+ZPressor(Overlap-based) | 21.49     | 0.727     | 0.194     |
> | 36        | DepthSplat+ZPressor(FPS-based)     | **23.88** | **0.815** | **0.150** |
>
> The results indicate that while the overlap-based anchor view selection strategy does yield some improvements, it does not consistently outperform the FPS-based strategy. We hypothesize this is due to the added constraints imposed by incorporating rotation information, which demands more precise camera poses. Given that our dataset only provides COLMAP-estimated poses, which may not be sufficiently accurate for such fine-grained angle considerations, opting for the default camera location proves to be a simpler and more effective approach. It is worth noting that the inclusion or exclusion of view angle information does not affect our core motivation and contribution. We will further investigate the overlap-based setting with more accurate pose data in future work.
>
> To further explore whether the Information Bottleneck (IB) principle could guide the anchor selection process, as suggested by the Reviewer 9obw, we designed an Adaptive Anchor View Selection strategy that adheres to this principle. Based on a given overlap threshold, the greedy algorithm stops when the overlap between all candidate views and the selected anchor views exceeds the threshold. This results in varying numbers of anchor views for different coverage regions, thus linking the information content of different scenes with the overlap between views, and ultimately enabling the selection of an appropriate bottleneck. The experimental results are as follows:
>
> | **Views** | **Methods**                        | **PSNR**  | **SSIM**  | **LPIPS** |
> | --------- | ---------------------------------- | --------- | --------- | --------- |
> | 36        | DepthSplat+ZPressor(Overlap-based) | 21.49     | 0.727     | 0.194     |
> | 36        | DepthSplat+ZPressor(Adaptive)      | **22.71** | **0.782** | **0.182** |
>
> ---
>
> ### **W3. MVSplat confidence-based pruning baseline**
>
> We appreciate the reviewer's insightful comment and suggestion regarding the use of confidence-based pruning for addressing "floater" artifacts. It's important to clarify that **floaters are a known characteristic** of feed-forward 3D Gaussian Splatting (3DGS) in dense view settings, as also observed in works like DepthSplat (refer to Figure B.3. in DepthSplat for similar observations). Our experiments are consistent with this behavior.
>
> While naively pruning Gaussians based on opacity (confidence) might seem intuitive, it is **generally sub-optimal and does not reliably mitigate floater artifacts**. This approach can inadvertently remove important Gaussians representing scene content, leading to "empty regions" or holes in the reconstruction, rather than exclusively targeting floaters.
>
> To illustrate this, we conducted experiments using confidence-based pruning as a baseline for compression (models are re-trained):
>
> | **Views** | **Methods**                                   | **PSNR**  | **SSIM**  | **LPIPS** |
> | --------- | --------------------------------------------- | --------- | --------- | --------- |
> | 24        | MVSplat                                       | 25.00     | 0.871     | 0.137     |
> | 24        | MVSplat+Confidence Pruning (prune_ratio=0.75) | 21.15     | 0.816     | 0.190     |
> | 24        | MVSplat+Confidence Pruning (prune_ratio=0.5)  | 26.94     | 0.886     | 0.130     |
> | 24        | MVSplat+ZPressor                              | **27.49** | **0.895** | **0.111** |
>
> *Note: For a fair comparison, the prune ratio for Confidence Pruning was set to 0.75 to yield a same number of Gaussians as ZPressor, and we also manually adjusted the prune ratio to 0.5 to improve performance.*
>
> 1. Numerically, confidence pruning offers some marginal improvement over the MVSplat baseline. However, its effectiveness is significantly lower than that of ZPressor. Visualizing the results reveals that **confidence pruning often introduces large empty regions**, particularly at object boundaries or in areas like the sky, which typically exhibit low confidence. This leads to significantly degraded visual quality. Furthermore, selecting an optimal pruning ratio for confidence-based methods is challenging.
> 2. This highlights the fundamental difference between the two approaches. Floater artifacts in feed-forward 3DGS often arise from inconsistent information across dense input views, causing the network to predict erroneous Gaussians in empty space. While pruning might remove some of these floaters, it does not address this root cause of redundancy at the prediction stage.
> 3. ZPressor, in contrast, focuses on compressing the multi-view features *before* 3D Gaussian prediction. By effectively reducing redundancy and focusing on salient information early in the pipeline, ZPressor directly tackles the "information overload" problem. This leads to a more robust and scalable model.
>
> We agree that exploring more sophisticated Gaussian pruning strategies post-prediction is a valuable direction for future research. We will include this ablation study and corresponding visualization results in the updated version.
>
> ---
>
> ### **Q1. Eq. 6: Cross-attention and Attention**
>
> We appreciate this clarification. We will correct Equation 6 and the accompanying text to reflect this precise terminology in the updated version.

---

> ### Author Response · Authors · 2025-08-04
>
> Dear Reviewer nH8F,
>
> Did we satisfactorily answer your questions? Would you like us to clarify anything further? Feel free to let us know, many thanks.
>
> Best regards,
>
> Authors of #3951

---

> > ### Comment · Reviewer_nH8F · 2025-08-06
> >
> > Thank you for the clarifications in the rebuttal. Reading the other reviews, it seems that other reviewers were also initially confused about the formulation and lack of mathematical rigor. I am overall happy with the responses, but would like to note that across all rebuttals, many changes to the paper have been promised, which hopefully will be carried out.
> > I will adjust my score.

---

> > > ### Author Response · Authors · 2025-08-06
> > >
> > > Dear Reviewer nH8F,
> > >
> > > Thank you for your response and for adjusting your score.
> > >
> > > We're glad that our rebuttal addressed your concerns regarding the formulation and mathematical rigor. We are fully committed to incorporating all the promised changes.
> > >
> > > Your feedback has been invaluable in helping us identify key areas for improvement, and we believe that the revised manuscript will be much stronger and easier for the community to understand.
> > >
> > > Best regards,
> > >
> > > Authors of #3951

---

### Official Review · Reviewer_9obw · 2025-07-02

**Clarity:** 2
**Significance:** 3
**Originality:** 3
**Rating:** 5
**Confidence:** 4

**Summary:**

This paper tackles the scalability problem of feed-forward 3DGS, which is limited by the encoder's capacity, leading to performance degradation or high memory use as the number of input views grows. The authors introduce ZPressor, a lightweight, architecture-agnostic module that efficiently compresses multi-view inputs into a compact latent state. This compression retains essential scene details while removing redundancies.

ZPressor splits the views into anchor and support sets. Anchor views serve as compressed states, and support views’ information is compressed into these anchors using cross-attention. Farthest point sampling is used to select anchor views, maximizing coverage with fewer views. Remaining views are assigned to the nearest anchor based on camera distance, and their features are fused into the anchors via cross-attention blocks.

**Questions:**

Pose-Free Settings:
Demonstrating the effectiveness of ZPressor in such configurations could strengthen the case for its versatility and robustness.

Clarification of Figure 6:
A deeper analysis of this phenomenon would help clarify the trade-off involved.

Adaptive Selection of Anchor Views:
If you could demonstrate how this adaptation might work, it could further enhance the practical utility of ZPressor.

**Ethical Concerns:**

["NO or VERY MINOR ethics concerns only"]

**Final Justification:**

The rebuttal addressed my main concerns. I have also read the reviews and rebuttals from the other reviewers. I would like to maintain my original opinion: This paper tackles the scalability issue of feed-forward 3DGS, which often leads to performance degradation and high memory consumption as the number of input views increases — a phenomenon consistently observed in many experiments. Tackling this challenge is valuable, and this paper represents a promising step in that direction. However, I also agree with the other reviewers that while the paper claims to be guided by IB theory, the current presentation lacks mathematical rigor. That said, the proposed approach is empirically consistent with the theory. I hope the authors can further clarify and strengthen their theoretical justification, using more precise language and derivations, in the camera-ready version.

**Limitations:**

yes

**Quality:**

3

**Strengths And Weaknesses:**

# Strengths

## Effectiveness:
ZPressor enables feed-forward 3DGS models to scale to over 100 input views at 480P resolution on an 80GB GPU. The results demonstrate that integrating ZPressor consistently improves the performance of baseline models, especially with a moderate number of input views (e.g., 12 views). It helps maintain reasonable accuracy and computational cost even with dense inputs (e.g., 36 views), where original models would typically degrade or run out of memory.

## Versatility:
ZPressor is architecture-agnostic, meaning it can be easily integrated into various feed-forward 3DGS models. The module improves performance under moderate input views and enhances robustness with dense view settings. It was successfully integrated into several state-of-the-art models (like PixelSplat, MVSplat, and DepthSplat) and tested on large-scale benchmarks (DL3DV-10K and RealEstate10K), consistently showing improvements.

## Insightful Experiments:
The paper provides valuable insight, particularly in Tables 1 and 2, where the performance of the state-of-the-art models drop as input views become denser. This highlights the importance of the proposed method in improving scalability.

# Weaknesses

## Lack of Pose-Free Settings:
The paper does not explore pose-free settings such as NoPoSplat, FLARE, or AnySplat, which could provide additional context to the proposed method’s versatility.

## Unexplained Performance Drop with Too Many Anchor Views:
Figure 6, which analyzes the bottleneck constraint, introduces an interesting idea. However, it doesn’t fully explain why performance drops when there are too many anchor views. While it's clear that too few anchor views lead to performance degradation due to over-compression, the impact of having too many anchor views is less clear. A more detailed explanation of this phenomenon would be beneficial.

##  Adaptive Selection of Anchor Views:
The method could benefit from a more flexible approach, allowing for adaptive selection of the number of anchor views based on input conditions. This could help optimize the performance further in different scenarios.

---

> ### Author Rebuttal · Authors · 2025-07-31
>
> ## **Response to Reviewer 9obw**
>
> We thank Reviewer 9obw for the positive assessment and constructive feedback. We are glad that the reviewer found our paper effective, versatile, and insightful. Below, we address the main discussion points raised by the reviewer:
>
> ---
>
> ### **Q1. Lack of pose-free settings**
>
> We agree that the pose-free setting is an important issue, but NoPoSplat, which based on the MASt3R [1] architecture and pre-trained weights, was only trained on two views, making it an unfair comparison to use as a baseline; FLARE does not have an official open-source implementation for NVS training; AnySplat was released on arXiv (May 29, 2025) after the NeurIPS submission deadline, so we chose the best options available to us in this submission.
>
> Following the reviewer's suggestion, we explored a **pose-free anchor view selection** method. We clustered the features of the input views with K-Means to group all views and then determined the anchor views. We conducted this experiment on DepthSplat:
>
> | **Views** | **Methods**                   | **PSNR**  | **SSIM**  | **LPIPS** |
> | --------- | ----------------------------- | --------- | --------- | --------- |
> | 36        | DepthSplat                    | 19.23     | 0.666     | 0.286     |
> | 36        | DepthSplat+ZPressor(w/o pose) | 22.81     | 0.791     | 0.174     |
> | 36        | DepthSplat+ZPressor(w/ pose)  | **23.88** | **0.815** | **0.150** |
>
> The experimental results demonstrate that ZPressor significantly outperforms the baseline, even with pose-free anchor selection, substantially improving the scalability of feed-forward 3DGS.
>
> ---
>
> ### **Q2. Clarification of figure 6: performance drop with too many anchor views**
>
> We hypothesize that when the scene's information content is relatively low (e.g., small camera baseline, similar views, as proxied by a small Context Gap like 50), a smaller number of anchor views (e.g., 7) is already sufficient to capture the essential scene information. Adding more anchor views in such a scenario might **introduce redundancy or ambiguity**, as these additional anchors might not observe genuinely new regions but rather re-observe already covered areas from slightly different perspectives. This could **lead to a less compact and potentially noisier latent representation**, hence the performance drop. In contrast, for scenes with higher information content (larger Context gap like 100), more anchor views are beneficial as they help cover more diverse perspectives and capture richer scene details. We will clarify this in the updated version.
>
> ---
>
> ### **Q3. Adaptive selection of anchor views**
>
> We agree that adaptive anchor selection is a promising direction. While our current farthest point sampling (FPS) method ensures a diverse and representative set of anchor views, we have explored an alternative following the reviewer’s constructive suggestion:
>
> We implemented an **adaptive anchor view selection method** based on the overlap of every input views. Based on a given overlap threshold, our greedy algorithm stops when the overlap of all candidate views with the selected anchor views is greater than the given threshold. This results in different numbers of anchor views for different coverage areas, which to a certain extent ensures that the amount of scene information remains within the ideal range and realizes adaptive anchor view selection.
>
> The experimental results of adaptive anchor view selection are as follows:
>
> | **Views** | **Methods**                             | **PSNR**  | **SSIM**  | **LPIPS** |
> | --------- | --------------------------------------- | --------- | --------- | --------- |
> | 36        | DepthSplat                              | 19.23     | 0.666     | 0.286     |
> | 36        | DepthSplat+ZPressor(Adaptive Selection) | 22.71     | 0.782     | 0.182     |
> | 36        | DepthSplat+ZPressor(FPS-based)          | **23.88** | **0.815** | **0.150** |
>
> The overlap-based adaptive anchor view selection, while showing some improvement, doesn't consistently beat the FPS-based strategy. We think this is because the overlap calculation requires highly accurate camera poses, which our COLMAP-estimated poses may not provide. Therefore, using the default camera location is simpler and more effective. The view angle information doesn't change our core motivation, and we'll explore overlap-based settings with more precise pose data in the future.
>
> We also counted the number of scenes in DL3DV-Benchmark(140 scenes) corresponding to different numbers of anchor views in this experiment when context gap is 50:
>
> | Number of Anchor Views | Number of Corresponding Scenes |
> | ---------------------- | ------------------------------ |
> | 6                      | 87                             |
> | 7                      | 23                             |
> | 8                      | 11                             |
> | 9                      | 19                             |
>
> The results show that the algorithm does indeed select different numbers of anchor views for different scenarios.
>
> ---
>
> ### Reference
>
> [1] Leroy, Vincent et al. “Grounding Image Matching in 3D with MASt3R.” ECCV 2024.

---

> > ### Author Response · Authors · 2025-08-04
> >
> > Dear Reviewer 9obw,
> >
> > Did we satisfactorily answer your questions? Would you like us to clarify anything further? Feel free to let us know, many thanks.
> >
> > Best regards,
> >
> > Authors of #3951

---

> > > ### Comment · Reviewer_9obw · 2025-08-06
> > >
> > > Thank you for the rebuttal — it addressed my main concerns. I have also read the reviews and rebuttals from the other reviewers. I would like to maintain my original opinion:
> > >
> > > This paper tackles the scalability issue of feed-forward 3DGS, which often leads to performance degradation and high memory consumption as the number of input views increases — a phenomenon consistently observed in many experiments. Tackling this challenge is valuable, and this paper represents a promising step in that direction.
> > >
> > > However, I also agree with the other reviewers that while the paper claims to be guided by Information Bottleneck (IB) theory, the current presentation lacks mathematical rigor. That said, the proposed approach is empirically consistent with the theory. I hope the authors can further clarify and strengthen their theoretical justification, using more precise language and derivations, in the camera-ready version.

---

> > > > ### Author Response · Authors · 2025-08-06
> > > >
> > > > Dear Reviewer 9obw,
> > > >
> > > > Thank you for your thoughtful feedback and for taking the time to read our rebuttal and the other reviewers' comments. We appreciate your positive assessment of our work and are encouraged that you see the value in our approach to address the scalability issues of feed-forward 3DGS.
> > > >
> > > > We agree that ZPressor should primarily be viewed as a practical technical module, while it is empirically consistent with IB theory, it is not our intension to claim that it is a mathematically rigorous application of IB theory. In our updated version, we will clarify this positioning, emphasize our technical contributions, and tone down the significance of IB theory.
> > > >
> > > > We have also provided additional explanations to Reviewer cRJd to further elaborate. We will integrate these discussions into the updated version to clarify our theoretical justification.
> > > >
> > > > Thank you again for your valuable comment.
> > > >
> > > > Best regards,
> > > >
> > > > Authors of #3951

---

### Decision · Program_Chairs · 2025-09-17

**Decision:**

Accept (poster)

**Comment:**

This paper introduces ZPressor, an architecture-agnostic module designed to improve the scalability of feed-forward 3D Gaussian Splatting (3DGS) models in dense-view settings. By employing a cross-attention–based compression mechanism inspired by the Information Bottleneck (IB) principle, ZPressor enables models to process more input views without excessive memory usage or performance degradation.

Reviewers agree that ZPressor addresses a relevant and challenging problem, demonstrates strong empirical effectiveness, and represents a meaningful contribution to dense-view 3DGS. Concerns about the rigor of the IB theory and certain experimental analyses are noted, but do not undermine the core technical contribution. Rebuttals clarified misunderstandings and provided additional context, increasing reviewers’ confidence in the paper.

Therefore, the AC recommends acceptance and suggests the authors: 1. Clarify the role and positioning of IB theory in the final version.
2. Enhance visualizations and explanations of technical claims, such as encoder capacity and dense-view behavior.